# Benefit–Risk Perceptions of FinTech Adoption for Sustainability from Bank Consumers' Perspective: The Moderating Role of Fear of COVID-19

Ruzita Abdul-Rahim [1,*] , Siti Aisah Bohari [1], Aini Aman [1] and Zainudin Awang [2]

1 Faculty of Economics and Management, Universiti Kebangsaan Malaysia (The National University of Malaysia), Bangi 43600, Selangor, Malaysia; p96999@siswa.ukm.edu.my (S.A.B.); aini@ukm.edu.my (A.A.)
2 Faculty of Business and Management, Universiti Sultan Zainal Abidin, Kuala Terengganu 21300, Terengganu, Malaysia; zainudinawang@unisza.edu.my
* Correspondence: ruzitaar@ukm.edu.my; Tel.: +60-389-215-764

**Abstract:** Industry 4.0 technologies, designed to optimize efficiencies, are indisputable change agents for sustainability. In the context of financial technology (FinTech), the burgeoning question concerns how to create FinTech natives from the COVID-19-pandemic-induced adoption and realize FinTech's impact on sustainability? Thus, this study had the following purposes: (1) to examine whether perceived benefits and risks affect FinTech services adoption; (2) to test the role of fear of COVID-19 in FinTech adoption; and (3) to investigate whether FinTech adoption contributes to sustainability. The hypotheses derived from the net valence framework, sustainable information society theory, and protection motivation theory were tested using structural equation modeling (SEM). Our online survey of bank consumers in Malaysia between December 2021 and February 2022 yielded 1279 usable questionnaires, randomly selected to generate 400 respondents. The results revealed that: (1) the perceived benefits significantly influence FinTech adoption, whereas perceived risk does not; (2) fear of COVID-19 moderates the perceived benefits–FinTech adoption relationship and fully mediates the perceived risk–FinTech adoption relationship; and (3) FinTech adoption significantly affects sustainability. This study demonstrates that FinTech adoption models must exploit consumer sentiment (e.g., fear) to optimize FinTech's benefits and risks, thereby creating FinTech natives to realize its impacts on economic, environmental, and social sustainability.

**Keywords:** consumer sentiment; FinTech adoption; FinTech natives; sustainability; fear of COVID-19; protection motivation theory; net valence framework; sustainable information society theory

## 1. Introduction

Industry 4.0 (I-4.0) technologies have caused tremendous disruptions to the financial services industry, as well as in manufacturing and other industries. These digital technologies are designed to improve production efficiencies, the key competitive advantage to survive modern, globalized markets. The combination of efficiency-driven property and digital structure makes I-4.0 technologies promising change agents for sustainability [1–7] and sustainable development [8–10]. They eliminate processes and resources that have caused irreversible environmental problems, resource depletion, and ecological imbalances [1]. To that effect, governments worldwide have acknowledged that I-4.0 technologies, including financial technology (FinTech), must be leveraged to recuperate environmental sustainability by de-materializing production and consumption, resulting in the significantly reduced use of natural resources [2,3,5]. The establishment of the United Nations Secretary General's Special Advocate for Inclusive Finance for Development (UN-SGSA) [4] affirms the contributions of FinTech services (henceforth, FinTech) to sustainable development [9,10]. FinTech contributes to financial inclusion by providing unbanked and underbanked consumers, especially low-income households and minority groups, access

to affordable and convenient financing to help increase their economic opportunities [4,8]. FinTech services reduce costs, enhance the quality of financial services, increase employment rates, reduce poverty by lowering transaction costs, facilitate everyday personal and professional life [5,6], and provide financial access through microfinance and crowdfunding [5]. Consumers' digital literacy and skills can also be enhanced through technology in financial services. FinTech services reduce energy consumption (e.g., fuel) and increase the protection of the environment (e.g., carbon emission) [1,6,7]. Although conceptually solid, in reality, these benefits cannot be realized because the adoption rates of FinTech services are low.

Realizing the impact on sustainability requires the generation FinTech natives (sustainable and mass adoption); however, consumers are still reluctant to embrace FinTech due to its controversies [11–16]. Consumers seem to consider the dangers and risks triggered by FinTech as more consequential than its benefits, which include conveniences, monetary savings, fast and seamless transactions, and economic efficiency [11,12,17–20]. FinTech is associated with cyber-related risks, broadly categorized into loss of privacy, compromised data security, rising financial losses due to frauds and scams, unclear legal status, lack of regulations, and risks that FinTech providers lack operational effectiveness [13]. Most of these risks are caused by the misuse and abuse of data, which has become more accessible in the digital universe. Despite the controversies, [16–19] argued that previous studies on FinTech have focused on its benefits. These authors [16–19] attributed the problem to the over-reliance on popular technology acceptance theories such as the technology acceptance model (TAM), theory of planned behavior (TPB), diffusion of innovations theory (DOI), and the unified theory of acceptance and use of technologies (UTAUT). Focusing on benefits could lead to suboptimal findings because the ubiquitous use of FinTech involves a complex trade-off between perceived benefits (returns/gains) and perceived risk (losses). It also does not justify that the cause of most financial problems bank consumers face in recent times is the failure to anticipate and manage risks and uncertainties [21]. This study addresses the gap in the literature by proposing the multidimensional benefit–risk perceptions in the net valence framework (NVF) to explain FinTech behavioral adoption during the pandemic.

To a great extent, the COVID-19 pandemic has presented a solid inducement for migrating to FinTech (and other I-4.0 technologies) [18,22–27]. Powered by financial technology (the origin of the term "FinTech") such as blockchain, Big Data, machine learning, and artificial intelligence (AI), FinTech has made it possible for consumers to perform financial transactions without the need for the physical presence of humans, money, or infrastructure. FinTech and I-4.0 technologies provide digital solutions to affected individuals, companies, and governments, thereby preventing the global economy from sinking into its worst depression. At the height of the COVID-19 pandemic, governments' responses to contain physical movement and promote safe physical contact induced a massive adoption of FinTech [18,27]. Without disqualifying the role of rules and regulation in shaping behavior, there are sufficient reasons to argue that users' behavioral shifts have also driven FinTech adoption during the pandemic [28]. COVID-19 is a highly contagious acute respiratory virus (SARS-CoV-2) transmittable through physical contact with infected humans or objects, including banknotes and coins [26,29]. Studies by [23–25] have shown that the fear of becoming infected has changed consumer spending and purchasing behavior toward online platforms. Before vaccines were available, an infection could lead to fatal consequences and caused many to suffer from morbid [25,28] and comorbid disorders [26]. In the FinTech context, As explained by [18], consumers suppress their concerns over FinTech risks to avoid COVID-19 infection. However, had [18] explicitly examined the fear of COVID-19, the result would have proven the role of consumer sentiment in technology adoption. The present study addresses the gap in the literature by proposing that the fear of COVID-19 explains FinTech behavioral adoption during the pandemic.

Malaysia represents an excellent context for this study because the landscape of its FinTech services industry is built on a cash-dominated economy. Its FinTech industry is characterized as a (regulated) open competition ground for incumbents and FinTech

startups (non-bank and digital banks). Before the COVID-19 outbreak, FinTech adoption in Malaysia was slow, despite the plan to transform the country into a cashless society [30]. The transformation is expected to result in cost savings worth 1% of the country's gross domestic product (GDP) [31,32]. Malaysia projected that technology-based innovations, specifically FinTech, would generate desired outcomes of financial inclusion: (i) convenient accessibility, (ii) high uptakes, (iii) responsible usage, and (iv) high satisfaction [32]. In their latest Financial Sector Blueprint 2022–2026 [31], Bank Negara Malaysia (BNM, the central bank of Malaysia) revised the e-payment per capita and compounded annual growth rate (CAGR) to more than 15% to speed up the digital transformation plan. At the same time, the country has invested in various initiatives to circumvent the FinTech threats. In 2020, the Malaysia Computer Emergency Response Team (MyCERT) recorded 10,790 cyber security incidents, up from 10,722 in 2019. Malaysia established MyCERT in 1997 to become the reference point for the internet community. MyCERT monitors the cyber security incidents in this country, classifying them into spam, malicious codes, intrusion, content-related, cyber harassment, fraud, vulnerabilities report, intrusion attempt, and denial of services. It also aids and advises the victims. Its web address is https://www.mycert.org.my/portal/statistics (accessed on 15 December 2021). MyCERT projects that the potential loss due to incidents from 2020 to 2024 could amount to MYR 51 billion, four times greater than the expected cost savings if its digital transformation plan materialized [31]. The significant potential for monetary loss and the dynamism of its FinTech industry present Malaysia as a good setting to examine how perceived risks influence FinTech adoption during times of crisis when FinTech solutions are most critical.

In light of the tremendous FinTech service uptakes due to COVID-19, the increasing cybercrime incidents, and the burgeoning sustainability issues, this study is a timely attempt to close the gaps in the literature by considering the antecedents of FinTech adoption from the perspectives of NVF [33], protection motivation theory (PMT) [34], and sustainable information society (SIS) theory [35]. Most previous studies on FinTech services have examined determinants of intention to adopt FinTech services [17–19,36–39]; however, the present study focuses on the experience of adopting FinTech during the pandemic, which corroborates the goal of changing the pandemic-induced behavior to loyalty [40]. This study addresses these gaps in the FinTech adoption literature through the following objectives:

(1) To examine whether perceived benefit and risk significantly influence FinTech behavioral adoption;
(2) To examine whether bank consumers' fear of COVID-19 moderates the relationship between perceived benefit and risk with FinTech behavioral adoption;
(3) To examine whether FinTech behavioral adoption contributes to sustainability.

Overall, this study contributes to the literature in the following ways. First, it addresses the gap in the literature on the impact of FinTech adoption on sustainability from the bank consumer perspective [41]. Although the COVID-19 pandemic has accelerated FinTech adoption, its implications for sustainability require the creation of FinTech natives. Second, it offers a context of a country undergoing a digital transformation while facing increasing cyber security incidents, which is therefore suitable to address the lack of empirical evidence on the effects of perceived risks [16–19]. As re-illustrated in Figure A1 (Appendix A), Malaysia's FinTech services industry is moving with the worldwide trend, reporting USD 27.75 billion in 2021 and expecting a CAGR of 22% to reach USD 74.38 billion by 2026. Previously, ref. [42] investigated FinTech adoption in Malaysia. However, their investigation of perceived risk was limited to internet banking, which is considered less complex and less risky than the novel FinTech services [12]. Third, this study takes a new twist in investigating the impact of the COVID-19 pandemic by proposing the role of fear of COVID-19 in influencing FinTech behavioral adoption. Fear of COVID-19 is a significant issue in the medical and psychiatric literature, because it has caused comorbid [26] and morbid disorders [25,28]. It has received considerable attention in financial market behavior studies, although this study is perhaps the first to examine its role in explicitly predicting technology adoption.

We achieved the study objectives by conducting an online survey on bank consumers between 4 December 2021 and 14 February 2022. As depicted in Figure A1 (Appendix A), digital payments represent the largest segment (54%) in the Malaysian FinTech market, with a total transaction value of USD 15.06 billion in 2021. Due to its dominance, this study targeted respondents among bank consumers who had experienced using FinTech to make payments or transfers through: (i) online banking, (ii) mobile banking, (iii) contactless debit/credit/prepaid cards, (iv) e-wallets, (v) online foreign exchange, and/or (vi) cryptocurrency e-wallets. These approaches yielded 1279 usable questionnaires that provided data from 400 randomly selected questionnaires. The data were tested using structural equation modeling (SEM). The results of this study are relevant to banks, the incumbents of the financial services industry that is being disrupted by FinTech startups. For FinTech companies, the results could help improve their FinTech adoption models, increasing their appeal to consumers. From the Malaysian policymaker's perspective, the results can help formulate the most effective strategies to accelerate FinTech adoption and realize the country's aspiration to become a cashless society and a regional leader in the digital economy.

The remainder of this paper is structured as follows. Section 2 reviews the literature related to the topic and develops the hypotheses. Section 3 describes the research methodology to examine the hypothesized models. The results and discussion are then presented. Section 5 concludes and presents the implications of the study.

## 2. Literature Review and Hypothesis Development

Financial technology (FinTech) may be a relatively new buzzword, but it has deep roots in the banking and financial sector. In a simple definition provided by [15], FinTech is a financial service that integrates finance and technology and is made available through advanced information and communications technology (ICT). The technology is behind automated teller machines (ATMs), credit cards, internet/online banking, and, more recently, mobile banking and e-wallets. The stretch of FinTech services reaches beyond online payment into financing through peer-to-peer (P2P) lending and crowdfunding, budgeting, financial planning, and investments [36]. These latest FinTech innovations are supported by combining old and new technologies such as blockchain, AI, machine learning, and Big Data, creating more complex and profound technologically enabled financial products and services [43]. By digitizing processes, FinTech has immense potential to solve various sustainability issues. The critical challenge lies in developing an effective FinTech adoption model [36] to create FinTech natives through a mass migration from the traditional financial services to garner the positive impact of FinTech services.

### 2.1. FinTech Behavioral Adoption in the Net Valence Framework

Developing a FinTech adoption model to cater to the circumstances described above requires a foundation based on an established technology adoption theory that considers factors driving and hindering the adoption. This study proposes the net valence framework (NVF) developed by [33] because, unlike most technology acceptance theories [16–19], it considers positive (benefits) and negative (risks) consequences. Evidence that perceived benefits, i.e., the positive aspects and utility [17,34,36], of technology influence technology adoption is empirically established. For FinTech services, the benefits include conveniences, monetary savings, fast and seamless transactions, and economic efficiency [11,12,15,17,19,20]. Considering the increasing cyber security incidents when FinTech adoption is accelerating, this study redirects the focus to the perceived risks, which refer to the subjective expectation of a possible loss [44]. Bauer [45] established perceived risk in his perceived risk theory, asserting that an individual's subjective risk assessments directly influence their decisions.

The NVF developed by [33] originates from economics and psychology disciplines [46]. The model is founded on an investigation of three types of consumer decision-making models: (i) a perceived return model in which consumers try to maximize expected positive

utility, (ii) a perceived risk model in which consumers try to minimize any negative utility, and (iii) a net valence model in which consumers try to maximize net returns or net valence, which is the difference between the expected positive and negative utilities. One study [33] found that the net valence model explains more variance in automobile brand preference than two other models. Hence, ref. [33] proposed that consumers perceive both positive and negative consequences of transactions but act to minimize negative utility, maximize positive utility, and maximize their overall net utility. The study became the foundation of NVF, which proposes that expected benefits must outweigh the expected risk for an endeavor to be viable. The NVF concept corresponds with financial and investment decisions, whereby an undertaking must be supported by expected returns higher than risks/costs.

NVF has prevailed as a solid theoretical basis in e-commerce studies [21,46,47]. As revealed by [47], perceived risks negatively influence e-commerce repurchasing intentions in South Korea. Similar results are documented from a sample of international respondents in repurchasing intentions using a cross-border e-commerce platform [46]. The findings by [21] reiterated the importance of risk consideration in financial decision-making, which is relevant to FinTech services given that cybercrimes are typically aimed at extorting the users' financial resources. In their study, ref. [21] discovered that most financial problems bank consumers currently face originate from neglect or failure to anticipate and manage the risks and uncertainties. Due to the escalating threats of its risks, FinTech is still considered a controversial financial management service. Although FinTech services provide immense benefits, bank consumers must assess the risks and costs. NVF captures this notion, that an individual intends to act if he perceives that the positive utility (perceived benefits) of the behavior (i.e., using a service) outweighs the associated negative utility (perceived risks) [33]. To examine this proposition, we first hypothesize that FinTech adoption is influenced by its perceived benefits, which refers to the consumers' perceptions of the positive value of using the services.

**Hypothesis 1 (H1).** *Perceived benefits positively influence FinTech behavioral adoption.*

As with any emerging technology, FinTech has been blamed for creating new types of risks and exacerbating the existing risks [48]. In the FinTech adoption literature, NVF has been used to explain near-field communication (NFC) mobile payment [17], biometric identity (ID) authentication for bank transactions [16], internet-only banks [20], and FinTech specifically [11,12,15,18]. The study by [17] extended the NVF with individual difference constructs to analyze restaurant consumers' intention to use NFC mobile payment in the United States. The results revealed that the perceived risk is insignificant, possibly due to the high confidence that the stricter regulations would ensure credit card providers are held accountable for any transaction errors or fraudulent payments. However, ref. [16] found that perceived concerns (similar to risks) significantly negatively influence the attitude toward biometric ID in banking transactions through ATMs in the same market. Another possible explanation for [17]'s results is that the sample mainly consists of younger males, who have the leniency to be indifferent toward risks. In South Korea, ref. [20] found that only two dimensions of perceived risk (i.e., functional risk and security risk) are significant. The study by [19] combined NVF with network externality theory to investigate the intention to use and continue to use internet-only banks.

Several studies have examined the multidimensional benefit–risk perspectives of FinTech adoption in South Korea [11,12] and Bahrain [15]. Some later studies [12,15] adopted the model developed by [11], which proposes perceived risk encompassing financial, legal, security, and operational risks. All studies [11,12,15] found that perceived risk is significantly negative in FinTech continuance adoption, despite the different respondents involved, i.e., consumers in [11,12] and bankers in [15]. Furthermore, ref. [12] revealed that risks of Fintech services, as compared with than standard internet-banking, are more pronounced because the former are more complicated and less predictable. A similar

result on perceived risk is documented by [36], who examined the influence of perceived value and risk and UTAUT factors on the adoption intention of FinTech internet wealth management platforms in China. The results in [36] show that perceived risk is the most influential factor in FinTech adoption intention.

Meanwhile, in investigating the intention to adopt FinTech services (INT) by a bank's users in China, ref. [37] used the extended TAM that places attitude (ATT) between the antecedents (including perceived risks) and INT. Their study found that ATT has a significantly positive effect in explaining INT, but the perceived risk is insignificant in influencing ATT. The effect of perceived risk on INT is most likely indirect because it significantly negatively impacts trust, which, in turn, significantly affects ATT. Stewart and Jürjens [13] investigated factors influencing the intention to adopt FinTech in Germany. The study found that data security (an element of FinTech risks) significantly positively influences the intention to adopt FinTech. The study by [18] integrated UTAUT with the extended NVF in examining factors affecting Jordanian citizens' intention to use FinTech. The study by [18] revealed that perceived risk does not directly influence the intention to use FinTech, but indirectly through trust, consistent with results in [37]. Similarly, ref. [19] found that perceived risk negatively affects Islamic FinTech adoption intention, but indirectly through perceived trust. The study by [19] integrated the perceived risk theory of [45], with perceived benefits and trust, to examine the relationships among users of Islamic banks in Pakistan.

Observations of the previous FinTech studies show that the research focuses on the intention to use the innovation [13,17–19,36–39] and continuance intention [11,12,15]. At the same time, there are limited studies on the actual use or experience of technology [40,41]. In a study on the intention to use blockchain-based cryptocurrency transactions among international users, ref. [38] pointed to the very low adoption as the main reason for not examining the actual usage of the technology. Given that users' intention does not automatically reflect on users' behavior, extending the extant literature by unearthing antecedents to the actual use of FinTech services is crucial. The focus shift from intention to behavior is timely because FinTech services have gained significant traction due to the COVID-19 pandemic. Based on the theoretical and empirical evidence, this study proposes the following hypothesis.

**Hypothesis 2 (H2).** *Perceived risk negatively influences FinTech behavioral adoption.*

### 2.2. FinTech Adoption and Sustainability from an SIS Perspective

During this critical time in history, when economic activities pose irreversible threats to the environment and natural resources, sustainability should be a crucial factor when considering a new technology [26,49], as is the case for promoting FinTech in Malaysia [30–32]. The Brundtland Commission Report by the United Nations World Commission on Environment and Development (UNWCED) defines sustainability: as "meeting the needs of the present without compromising the ability of future generations to meet their needs" [50] (p. 41). It concerns sustainability (concerning development), renewability (regarding resources), and sustained growth [50]. Instilling sustainability in an information society requires "a holistic and integrated policy framework of environmental compatibility, economic stability, social sustainability, and cultural diversity" [35] (p. 7). The use of technology could also lead to income growth and cost reduction, facilitating everyday personal and professional life and increasing satisfaction with online products [6,40]. It also contributes to ecological sustainability by de-materializing production and consumption, thereby reducing the use of natural resources [2,49]. However, as suggested by [3], sustainable long-term policies are necessary to make the pandemic-induced FinTech adoption permanent for the country, creating FinTech natives in order to realize economic, environmental, and social sustainability.

FinTech is a relatively new area; therefore, this study supports its arguments by including studies that linked ICT adoption and sustainability. Of particular interest is [6], who introduced the sustainable information society (SIS) theory to depict various dimen-

sions shaping ICT adoption and their impacts on different forms of sustainability. In a study involving Polish households, ref. [6] referred to ICT as digital household systems that have moved beyond the generic concept of ICT, such as e-health, e-commerce, e-government, e-shopping, e-education, and e-working. The study by [6] proposed that household sustainability consists of ecological, economic, socio-cultural, and political sustainability. The relationship between technology adoption and sustainability in the present study is consistent with the SIS theory. However, this study views sustainability from a FinTech context, primarily aiming to increase access to financial services for underbanked and unbanked consumers.

Empirically, the contribution of FinTech to sustainability has been established, but mainly addressing the supply side (i.e., firms and countries). Integrating dynamic capability views and contingency theory, ref. [7] found that FinTech (specifically, Big Data and predictive analysis) significantly impacts social and environmental sustainability in supply chains. Another study by [50] found similar results in FinTech P2P lending adoption, which leads to the sustainability of small food businesses in Indonesia. Consistent with the technological knowledge spillover theory, ref. [9] found that FinTech development improves the sustainable performance of 59 healthcare firms in 11 Asia-Pacific countries. Meanwhile, ref. [1] discovered a U-shaped relationship between FinTech (P2P platforms) and sustainable development in China on a macroeconomic level. Similarly, ref. [10] revealed that FinTech services are an effective prompter for sustainable development across all financial and non-financial industries in Korea.

Within the scant literature on FinTech at the demand-side (i.e., consumer or household), ref. [41] found that mobile money services have improved financial inclusion and positively impacted the low-end segment of the population in Uganda. Mobile money services are a form of FinTech that allow an individual with a mobile phone to set up a mobile money account with the mobile network operator and deposit cash in exchange for electronic money [51]. Mobile money positively impacts sustainability through the United Nations (UN) Sustainable Development Goal (SDG) covering Gender Equality (SDG5), SDG 8—Decent Work and Economic Growth and expanding financial inclusion through mobile money, and SDG 10—Reduce Inequalities [41]. Chikalipah [52] conducted a similar study among low-income households in Zambia, and found strong evidence showing that mobile money users mainly used the service for money transfers. Had they used mobile money for savings and paying credit balances, they could have improved their consumption through borrowing, reduced their vulnerability to shocks through risk diversification, and increased investment through savings. The study by [52] emphasized that it is through these channels that mobile money services can contribute to achieving the SDGs. Similarly, ref. [51] found that a mobile money cash transfer program in Niger has improved household diet diversity and intra-household bargaining power for women, because the FinTech services address key logistic challenges in cash transfers. Suri et al. [53] examined the acceptance of M-Shwari, one of the world's most popular digital loan service, in Kenya. The study by [53] found that 34% of eligible households used the loan, which has improved their financial access and resilience. These findings resonate with the arguments that FinTech-based financing platforms such as P2P lending and crowdfunding are capable of driving financial inclusion [48]. Leveraging FinTech services for financial inclusion would generate more substantial effects in developing markets [54] because they bridge the gaps for unserved and underserved people in traditional financial services. The following hypothesis assesses whether FinTech adoption among bank consumers in Malaysia contributes to sustainability.

**Hypothesis 3 (H3).** *FinTech behavioral adoption positively impacts sustainability.*

*2.3. FinTech Adoption and Fear of COVID-19 from a PMT Perspective*

When the world was shocked by the COVID-19 outbreak in Wuhan, China, which was later declared a pandemic, many were reluctant to accept the gravity of COVID-19. Within a few months, governments worldwide were forced to enforce lockdowns or physical

containment measures as the world was hit harder by second and third waves of infections. These relapses have created fear among the public to the extent of experiencing comorbid disorders [26], stressing the possibility that the world might never be free from COVID-19 [55]. The consumer behavior literature defines fear as the negative consequences of a specific event that can change consumer behavior and attitude, consistent with the conceptualization by the protection motivation theory (PMT) proposed by [34]. The theory posits that behavioral responses result from various dimensions of fear evaluations [34]. PMT has been adopted to explain human behavior during the pandemic [23]. The fear of COVID-19 contagion has become an emerging issue [28]; thus, it is imperative to learn how it influences consumer spending and purchase behavior [23,24]. Some studies [25,28] have indicated that the COVID-19 pandemic has caused morbid disorders, and consumers increasingly purchase products and services through online platforms due to the perceived safety offered by the internet and online technologies [34].

PMT applies to the change in bank consumers' behavior due to COVID-19, including their susceptibility to FinTech services. As suggested by [18], consumers' concerns about FinTech risks are masked by their fear of COVID-19 infection. The fear encompasses the virus's transmissibility through cash (banknotes and coins). The European Central Bank responded to this public concern by conducting a study on the survivability of COVID-19 on their banknotes. The study concludes that "as for many other similar viruses, SARS-CoV-2 survives on banknotes and coins for 30 min to a maximum of several days (significantly shorter than on door handles, for instance), but only in limited quantities" [29] (p. 15). The unprecedented public concern has led central banks in countries such as China, South Korea, Hungary, and Kuwait to implement measures to sterilize or quarantine banknotes to ensure that cash leaving their currency centers does not carry viruses [26]. The fear of cash-carrying-COVID-19 should have been more prominent in economies where cash is dominant. The central banks in India, Indonesia, Georgia, Malaysia, and several other countries have encouraged cashless payments [26].

In a comprehensive study involving 74 countries, ref. [27] revealed that the spread of COVID-19 and related government lockdowns have led to an average daily increase of roughly 5.2 to 6.3 million finance mobile application downloads. That is an increase of about 316 million app downloads since the pandemic outbreak, considering prior trends from October 2019 to April 2020. As asserted by [55], in many countries, bank consumers had increased their usage of FinTech services (e.g., online banking, mobile banking, contactless card payment, and e-wallet) to avoid physical contact with objects, including cash touched by multiple persons. A study by [55] discovered that research and education industry respondents in Bulgaria have significantly adopted more FinTech services during the pandemic, i.e., from March to May 2020. As described in the Introduction, Malaysia has experienced a similar trend. Thus, this study proposes that the fear of COVID-19 moderates the relationship between perceived benefits and perceived risk with FinTech behavioral adoption.

**Hypothesis 4a (H4a).** *Fear of COVID-19 moderates the relationship between perceived benefit and FinTech behavioral adoption.*

**Hypothesis 4b (H4b).** *Fear of COVID-19 moderates the relationship between perceived risk and FinTech behavioral adoption.*

*2.4. The Conceptual Framework*

This study draws from the theoretical and empirical arguments discussed in previous sections to develop an integrated model between the net valence framework (NVF) and sustainable information society (SIS) theory, making the ultimate dependent variable sustainability. In addition, this study incorporates the protection motivation theory (PMT) by introducing the moderating effect of fear of COVID-19. The proposed research model is depicted in Figure 1.

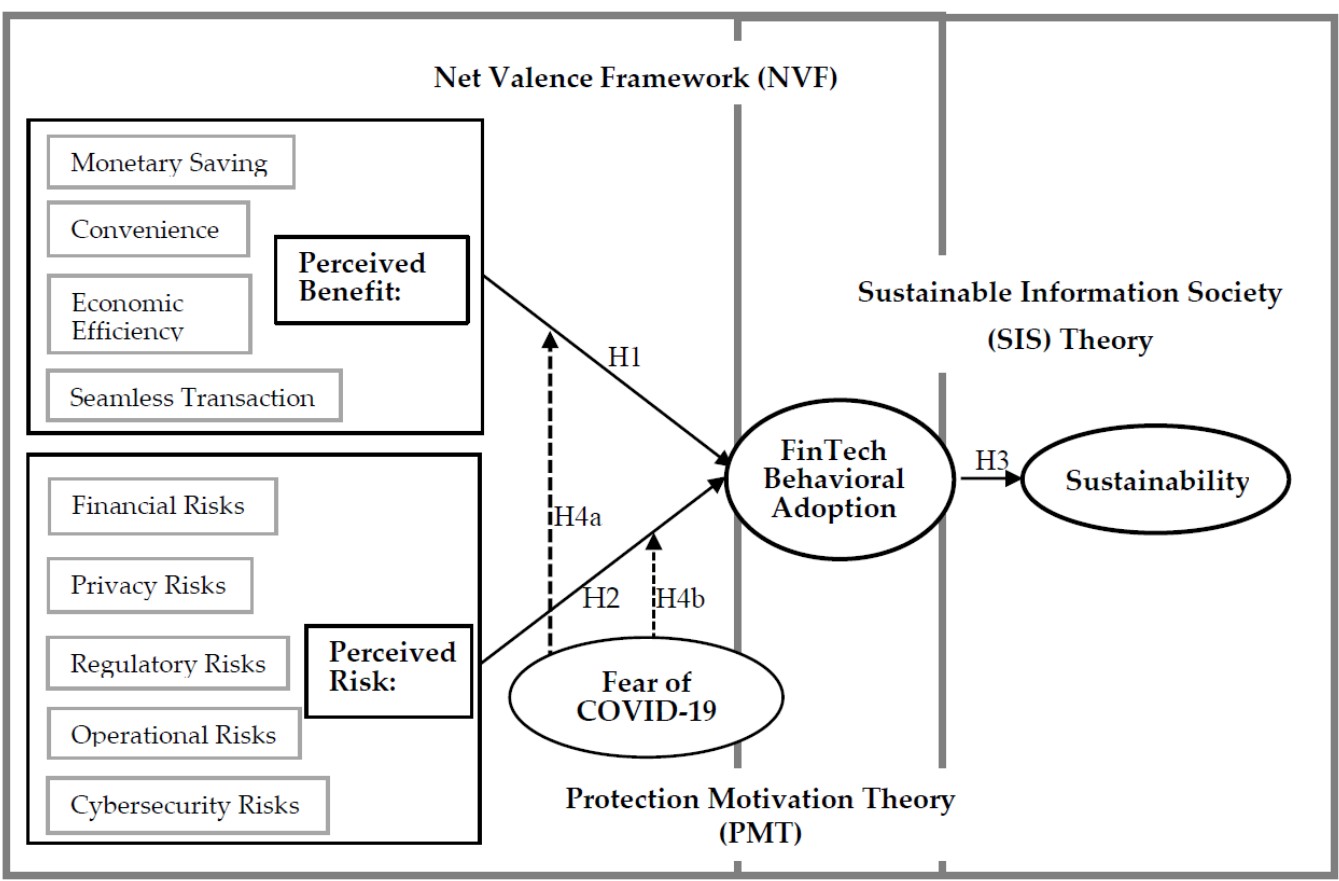

**Figure 1.** Proposed research framework.

Following [56], this study tests the perceived benefit and perceived risk as second-order constructs to reduce measurement issues. As emphasized by [56], the goals of measurement are to ensure that: (i) it reflects all key aspects of the conceptual definition; (ii) the items do not include irrelevant things that are not part of the conceptual domain; and (iii) the items are clear and explicit. In a study that employed NVF constructs, ref. [57] assessed the effect of perceived benefit and perceived risk by modeling the two variables as first-order and second-order constructs. The results reveal that NVF with perceived benefits and risks as second-order constructs explains the most significant variance in continuance intention to use social networking sites (SNSs). Subsequently, this study examines perceived benefits and risks as second-order constructs to avoid measurement issues.

## 3. Methodology

This study employed a quantitative research design using a survey method that collected data by administering questionnaires. The following sub-sections describe the survey instruments and sampling methods for selecting the final respondents. The final sub-section explains the methods used to analyze the data and test the hypotheses.

### 3.1. Designing the Survey Instrument

This study developed the FinTech adoption survey instrument by adapting constructs and items from previous studies in technology adoption, the usage of innovations, sustainability, and the fear of COVID-19. Details of these constructs are depicted in Figure 1, with their items and respective sources provided in Table A1 (Appendix A). The foundation of the FinTech adoption instrument is built on the NVF to ensure that the adoption decision considers a positive net valence of perceived benefit and perceived risk. Perceived benefit is initially built on four dimensions: convenience, monetary saving, seamless transaction, and economic efficiency [11,12,17,20]. Meanwhile, perceived risk comprises five dimen-

sions: financial, regulatory, cybersecurity, privacy, and operational risk [11,12,17,46]. We adapted [6]'s measurement to tap three dimensions of sustainability: economic sustainability, social sustainability, and environmental sustainability. FinTech behavioral adoption was adapted from [58,59], and the fear of COVID-19 was adapted from [28]. The item statements were measured using a 10-point Likert scale, ranging from 1 (strongly disagree) to 10 (strongly agree). The 10-point scale is more efficient and gives higher discriminant and convergent validity than 5- or 7-point scales. The 10-point Likert scale provides more freedom of choice to the respondents where the forced measure would not occur [60]. Previous studies, especially those employing SEM, have also used a 10-point Likert scale [60,61].

The instrument underwent several stringent procedures before being administered in the field study [60,61]. First, we conducted pre-testing on the instrument to obtain expert validation of its content, face, and criterion validity. The experts were academics (two university professors) and a practitioner (a senior executive from the Securities Commission of Malaysia, the sole financial market regulator). The items were refined according to the experts' feedback. Second, the refined questionnaires were administered to 100 respondents in the pilot study stage. We employed the exploratory factor analysis (EFA) procedure to assess the usefulness of the measurement items and determine their factorial structure. Based on the EFA results, we removed several items with poor factor loading and rearranged useful items for the field study questionnaire. The perceived benefit items emerged as two sub-constructs, namely, monetary and non-monetary benefits. Meanwhile, perceived risk items emerged as three sub-constructs: personal data protection and regulatory and financial risks. The final instrument's items are presented in Table A1 (Appendix A).

### 3.2. Data Analytical Methods

To analyze the field study data, we first employed confirmatory factor analysis (CFA) to test the latent constructs before performing SEM for hypothesis testing [57]. CFA assesses the instrument's validity using four tests: construct validity, convergent validity, discriminant validity, and composite reliability. We also evaluated the normality distribution and common method variance. The normality assessment is based on every item's skewness and kurtosis. Data are normally distributed if the skewness falls within the $-1.0$ to $1.0$ range. For a sample size greater than 200, skewness values between $-1.5$ and $1.5$ and kurtosis values between $-3.0$ and $3$ are acceptable [62]. This study used Harman's one-factor solution to assess the common method variance and extract a single factor accountable for variance in the dataset [57]. Finally, once the validation procedures were completed and the results met the requirement for parametric analysis, this study developed the structural models. The SEM procedure was then performed to estimate the parameters in the structural model for testing the proposed hypotheses. SEM path analysis is employed to test the direct effect and mediating effect in structural models. The estimated models must fulfill three goodness-of-fit criteria: absolute fit (root-mean-square error of approximation, RMSEA < 0.08), incremental fit (comparative fit index, CFI > 0.9), and parsimonious fit (chi-square/df < 5.0). We used the multi-group CFA (MCFA) to estimate the moderating effect. The CFA, MFCA, and SEM path analyses were carried out using IBM-SPSS-AMOS 24.0.

### 3.3. Sampling for the Field Study

The target population of this study is bank consumers, considering that banks are the incumbents in the financial services industry that FinTech has disrupted. The target respondents are Malaysian aged 18 and above, who own a bank account, live in Malaysia, and have experience of making money transfers or payments via any of the following FinTech services: (i) internet banking, (ii) mobile banking apps, (iii) contactless debit/credit/prepaid card, (iv) e-wallets, (v) online foreign exchange, and (vi) cryptocurrency e-wallets. In Malaysia, 92% of the adult population have active bank accounts, and this percentage reached approximately 95% by 2020 [32]. However, as recorded in Statista, the FinTech industry in Malaysia is still growing. In 2021, after COVID-19 accelerated its adoption, the FinTech service adoption rate is still around 50% (15.96 million) of the population. This slow

progress could hinder the country's aspiration to become a cashless society by 2025 [30] and the regional leader in the digital economy by 2030 [31].

We developed the study sampling frame by distributing the questionnaires online using purposive and snowball sampling methods to gather as many responses as possible. The online data collection, administered from 4 December 2021 to 14 February 2022, gathered 1368 responses. After data cleaning procedures, the sample frame consisted of 1279 usable questionnaires consecutively numbered 1 to 1279 based on the time of response submission. This study extracted 400 questionnaires using a simple random sampling method for further analysis. Using Monte Carlo data simulation techniques, ref. [63] suggests that a sample of 30 to 460 cases is acceptable for SEM analysis. We limited the sample to 400 respondents; [60] indicated that a sample larger than 400 respondents would cause SEM to become sensitive, causing any difference to be detected and the goodness-of-fit measures to exhibit poor fit [60].

## 4. Results

### 4.1. Profile of Respondents

Table 1 shows the demographic representation of the 400 respondents. The majority of them were female, aged between 22 and 25 years old, had a bachelor's degree as their highest qualification, were students, and earned a monthly income of MYR 1000 or less. Interestingly, although FinTech adoption was at its peak during the pandemic, its uses in Malaysia are still for basic services, specifically internet/online banking, mobile banking, e-wallets, and contactless debit/credit/prepaid cards. Advanced FinTech services such as online foreign exchange and cryptocurrency e-wallets are still low in use, and were thus excluded from this analysis.

**Table 1.** Demographic characteristics of the respondents.

| Demographic/Characteristics | Frequency | % | Demographic/Characteristics | Frequency | % |
|---|---|---|---|---|---|
| Gender: | | | Age: | | |
| Male | 129 | 32.3 | 18–21 years | 95 | 23.8 |
| Female | 271 | 67.8 | 22–25 years | 194 | 48.5 |
| Total | 400 | 100 | 26–29 years | 27 | 6.8 |
| | | | 30–34 years | 28 | 7.0 |
| Highest Education Level: | | | 35–39 years | 16 | 4.0 |
| SPM/STPM/STAM [a] | 33 | 8.4 | 40–49 years | 23 | 5.8 |
| A-level/Foundation/Matriculation | 4 | 1.2 | 50 years and older | 17 | 4.3 |
| Diploma | 34 | 4.6 | Total | 400 | 100 |
| Bachelor | 259 | 64.8 | | | |
| Master and Professional Certificate | 70 | 17.6 | Main Profession Category: | | |
| Total | 400 | 100 | Professionals in the Tech field [b] | 34 | 8.5 |
| | | | Other Professionals [c] | 58 | 14.5 |
| Total monthly income: | | | Management | 31 | 7.8 |
| MYR 1000 or less | 237 | 59.3 | Technical staff and technician | 6 | 1.5 |
| MYR 1001 to MYR 3000 | 68 | 17.0 | Front line employees | 3 | 0.8 |
| MYR 3001 to MYR 5000 | 45 | 11.3 | Business owner (online) | 4 | 1.0 |
| MYR 5001 to MYR 8000 | 31 | 7.8 | Business owner (offline) | 5 | 1.3 |
| MYR 8001 to MYR 10,000 | 7 | 1.8 | Student | 247 | 61.8 |
| More than MYR 10,000 | 12 | 3.0 | Do not work | 12 | 3.0 |
| Total | 400 | 100 | Total | 400 | 100 |

Notes: Description for superscripts: [a], equivalent to a high-school diploma; [b], information technology (IT) engineers, scientists, and software engineers; and [c], non-technical professions such as physicians, doctors, lawyers, accountants, and non-IT scientists.

### 4.2. Results of the CFA Procedures

Table 2 presents the CFA results, specifying the factor loading for dimensions and items under each construct and sub-construct. It also reports the average variance extracted (AVE) and composite reliability (CR) for each construct. The table shows that a few items were removed due to poor factor loading. The items with factor loadings above 0.6 [60] were retained to satisfy the unidimensionality criterion. An exception is item FC3, which is above 0.5 [64]. The results of AVE for the five constructs under study satisfied the convergent

validity criterion (AVE > 0.5). Similarly, the results of CR for the five constructs met the composite reliability criterion (CR > 0.6).

**Table 2.** Standardized loadings, AVE, and CR for each construct, sub-construct, and item.

| Construct (Abbreviation) | Sub-Construct/Item-Label | Factor Loading | AVE (above 0.5) | CR (above 0.6) |
|---|---|---|---|---|
| (1) Perceived Benefits (PBEN) | Non-monetary benefits | 0.869 | 0.727 | 0.842 |
| | Monetary benefits | 0.836 | | |
| | NB1 | 0.975 | | |
| | NB2 | 0.976 | | |
| | NB3 | 0.982 | | |
| | NB4 | 0.945 | | |
| Non-monetary benefits (NBs) | NB5 | 0.962 | 0.935 | 0.992 |
| | NB6 | 0.952 | | |
| | NB7 | 0.982 | | |
| | NB8 | 0.977 | | |
| | NB9 | 0.951 | | |
| | MB1 | 0.847 | | |
| Monetary benefits (MBs) | MB2 | 0.927 | 0.818 | 0.931 |
| | MB3 | 0.937 | | |
| (2) Perceived Risk (PRISK) | Personal Data Protection Risk | 0.832 | 0.619 | 0.829 |
| | Regulatory Risk | 0.747 | | |
| | Financial Risk | 0.778 | | |
| | PR1 | 0.839 | | |
| | PR2 | removed | | |
| | PR3 | 0.888 | | |
| | PR4 | 0.881 | | |
| | PR5 | 0.921 | | |
| Personal Data Protection Risk (PR) | PR6 | 0.932 | 0.806 | 0.974 |
| | PR7 | 0.920 | | |
| | PR8 | 0.925 | | |
| | PR9 | 0.931 | | |
| | PR10 | removed | | |
| | PR11 | removed | | |
| | PR12 | 0.839 | | |
| | RR1 | 0.920 | | |
| Regulatory Risk (RR) | RR2 | 0.967 | 0.897 | 0.972 |
| | RR3 | 0.954 | | |
| | RR4 | 0.947 | | |
| | FR1 | 0.904 | | |
| Financial Risk (FR) | FR2 | 0.945 | 0.874 | 0.965 |
| | FR3 | 0.946 | | |
| | FR4 | 0.944 | | |
| (3) FinTech Behavioral Adoption of Basic FinTech Services (FAB) | FAB1 | 0.780 | 0.547 | 0.828 |
| | FAB2 | 0.745 | | |
| | FAB3 | 0.755 | | |
| | FAB4 | 0.673 | | |
| Adoption of Advanced FinTech Services (FAA) | FAA1 | removed | | |
| | FAA2 | removed | | |
| | FC1 | 0.881 | | |
| | FC2 | 0.859 | | |
| | FC3 | 0.565 | | |
| (4) Fear of COVID-19 (FC) | FC4 | 0.848 | 0.692 | 0.939 |
| | FC5 | 0.844 | | |
| | FC6 | 0.920 | | |
| | FC7 | 0.857 | | |
| | S1 | 0.901 | | |
| | S2 | removed | | |
| | S3 | 0.948 | | |
| | S4 | 0.860 | | |
| (5) Sustainability (S) | S5 | 0.879 | 0.804 | 0.953 |
| | S6 | 0.892 | | |
| | S7 | removed | | |
| | S8 | removed | | |
| | S9 | removed | | |

This study used Fornell and Larcker's criterion to assess the discriminant validity of the model. As shown in Table 3, all the pair-wise construct correlation values (figures

in diagonal cells) are lower than the square root function of AVE, thus satisfying the discriminant validity criterion.

**Table 3.** Discriminant validity summary for constructs.

| Construct | Perceived Benefits | Perceived Risk | FinTech Behavioral Adoption | Fear of COVID-19 | Sustainability |
|---|---|---|---|---|---|
| Perceived Benefits | 0.853 | | | | |
| Perceived Risk | 0.502 | 0.787 | | | |
| FinTech Behavioral Adoption | 0.678 | 0.329 | 0.740 | | |
| Fear of COVID-19 | 0.429 | 0.467 | 0.468 | 0.832 | |
| Sustainability | 0.762 | 0.466 | 0.698 | 0.484 | 0.897 |

Note: Values in diagonal cells are the square root of the AVE of the respective construct.

In addition, the skewness and kurtosis values are consistently in the range of −1.251 to 0.464 and −0.649 to 1.225, respectively. These values indicate that the data were normally distributed and met the assumption of parametric statistical analysis [62]. Meanwhile, Harman's one-factor solution confirmed that a single factor explains 44.07%, i.e., less than 50% of the total variance [57]. The result indicates that the procedure is free from method bias. All assessments for latent constructs satisfied the requirements of parametric statistical analysis. This study proceeded with finalizing the structural model and performed the SEM procedure to estimate the regression parameters in the model.

### 4.3. Results of SEM Path Analysis

SEM generates a set of fitness indexes for the estimation model to indicate the construct validity. The results in Figure 2 confirm the construct validity, because the fitness indexes have fulfilled three model-fit criteria, namely, absolute fit (RMSEA < 0.08), incremental Fit (CFI > 0.9), and parsimonious fit (chi-square/df < 5.0).

The $R^2$ value in Figure 2 indicates that perceived benefits and perceived risks can explain 59% of the variations in FinTech behavioral adoption. At the same time, the second $R^2$ value suggests that 60% of variations in sustainability could be explained by FinTech behavioral adoption. Based on the $R^2$ values, it can be concluded that the models explain sufficient variations in sustainability. In other words, the two exogenous constructs in the model adequately explain sustainability which is achievable through FinTech adoption.

Figure 2 shows the standardized estimates of the paths; Table 4 shows the unstandardized estimates, complete with their significance values. The path coefficient estimates reveal that perceived benefit positively and significantly influenced ($p < 0.01$) FinTech behavioral adoption during this pandemic crisis. This study concludes that the H1 and H3 hypotheses are supported. Meanwhile, H2 is rejected because the perceived risk is insignificant in explaining FinTech behavioral adoption ($p > 0.10$).

**Table 4.** The unstandardized regression path coefficients of constructs and significant values.

| Construct | Path | Construct | Estimate | SE. | CR. | Prob. | Result |
|---|---|---|---|---|---|---|---|
| Perceived Benefits | → | FinTech Behavioral Adoption | 0.693 | 0.061 | 11.357 | *** | Significant |
| Perceived Risk | → | FinTech Behavioral Adoption | 0.023 | 0.055 | 0.419 | 0.676 | Not Significant |
| FinTech Behavioral Adoption | → | Sustainability | 0.856 | 0.059 | 14.462 | *** | Significant |

Note: SE, standard error; CR, critical ratio; prob, probability. Asterisks *** correspond to significance level at 1%.

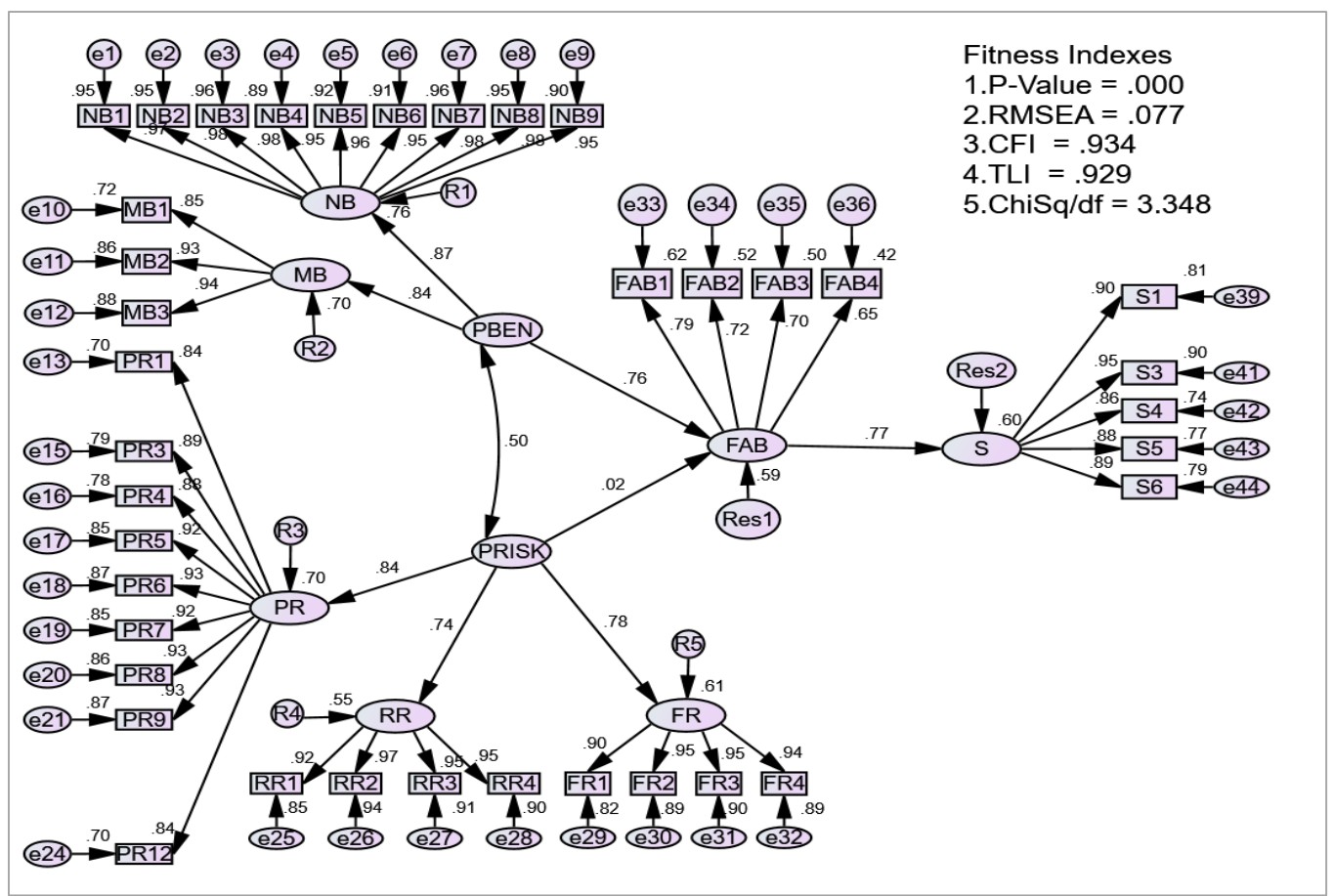

**Figure 2.** Standardized path coefficients between main constructs in the structural Model. Note: All abbreviations are defined in Table 2.

### 4.4. Results of MCFA on Moderator Effect of the Fear of COVID-19

Following [49], we employed multi-group CFA (MCFA) to test the moderating effect of fear of COVID-19 on initial relationships. The MCFA procedure requires the data of a moderator to be grouped into two levels. The moderation test is conducted by computing the chi-squared difference between both models and datasets. The hypothesis is supported if the chi-squared difference between the two models is greater than 3.84 for at least one of the two datasets [49]. In this study, we divided our data into low fear and high fear of COVID-19 datasets in both constrained and unconstrained models to estimate the chi-squared and degree of freedom (df) values.

The results in Table 5 show that both chi-square differences are higher than 3.84. Therefore, this study concludes that a fear of COVID-19 moderates the relationship between perceived benefit and FinTech behavioral adoption. Thus, H4a is supported. Another moderator hypothesis (H4b) could not be tested because the direct effect of perceived risk on FinTech behavioral adoption is not supported [49].

**Table 5.** The moderator test for low and high fear of COVID-19 group data.

| Level of COVID-19 Fear | Tests | Constrained Model | Unconstrained Model | Chi-Square Difference | Result on Moderation |
|---|---|---|---|---|---|
| Low-fear group | Chi-square | 1814.570 | 1810.295 | 4.28 | Significant |
| | df | 660 | 659 | 1 | |
| High-fear group | Chi-square | 1603.672 | 1563.507 | 40.17 | Significant |
| | df | 660 | 659 | 1 | |

Notes: Constrained vs. unconstrained models.

### 4.5. Results of SEM Path Analysis on Mediation Effect of Fear of COVID-19

The finding that consumers' perceptions about FinTech risks are not crucial in their adoption decision is concerning. It suggests that consumers are unaware of the threats that FinTech services can implicate for them, despite the increasing cases of FinTech-related cybercrimes. It is also possible that consumers consider the risk benign and not able to cause severe damage to the extent that it prevents them from adopting FinTech. This study recognized the need to rigorously examine the role of perceived risk to address this issue. It models the fear of COVID-19 as a mediator linking perceived risk and FinTech behavioral adoption. The graphical results are displayed in Figure 3.

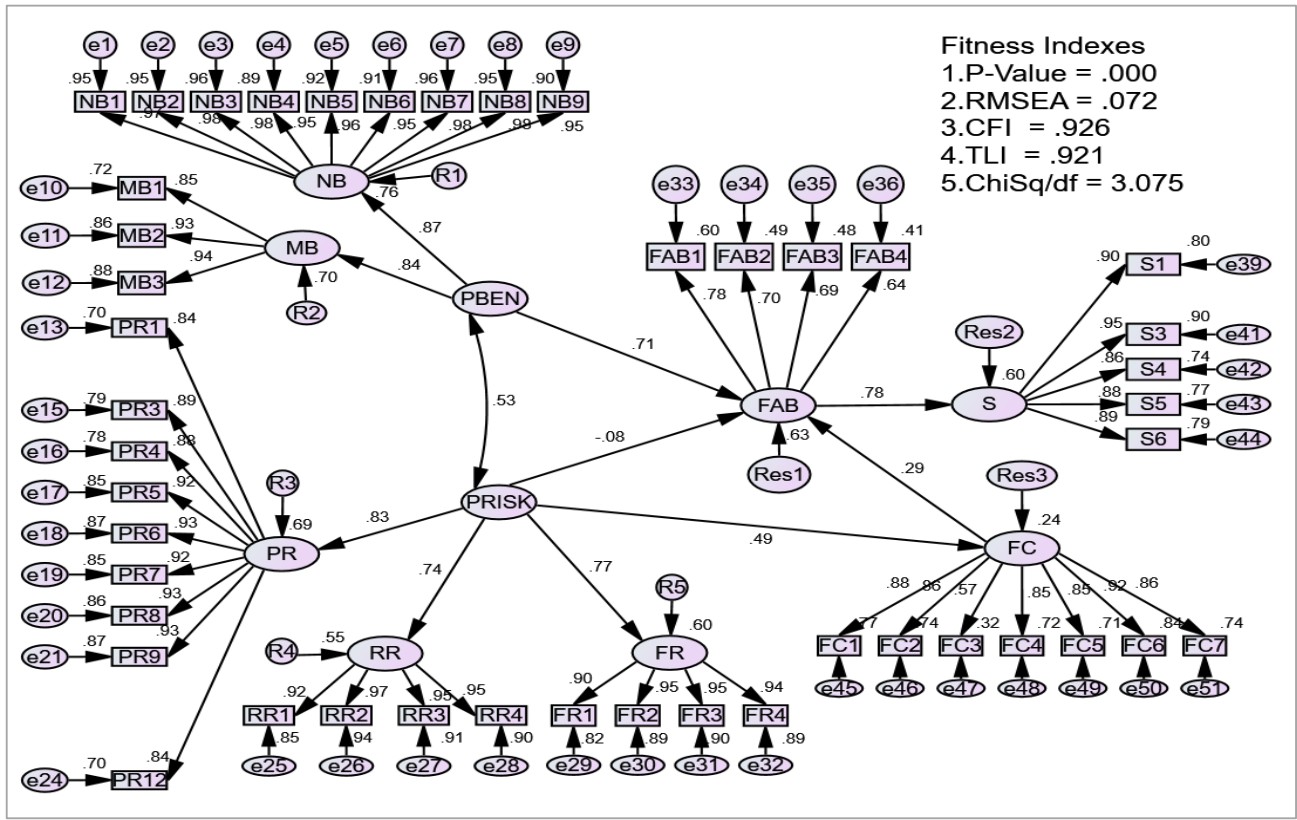

**Figure 3.** Path coefficients of the main relationships with a mediator in the structural model. Note: All abbreviations are defined in Table 2.

The text output in Table 6 reveals that the direct effect of perceived risk on FinTech adoption, as hypothesized in H2, is not significant. However, perceived risk significantly influences the fear of COVID-19, which, in turn, has a significant influence on FinTech behavioral adoption. These results suggest that perceived risk indirectly influences FinTech behavioral adoption, and the magnitude of the indirect effect is 0.1434 (i.e., (0.491 × 0.292)). The indirect relationship is fully mediated by fear of COVID-19.

**Table 6.** Fear of COVID-19 as a mediator in the relationship between perceived risk and FinTech behavioral adoption.

| Construct | Path | Construct | Estimate (β) | Std. Estimate (β) | Prob. | Result |
|---|---|---|---|---|---|---|
| Perceived Risk | → | Fear of COVID-19 | 0.640 | 0.491 | *** | Significant |
| Fear of COVID-19 | → | FinTech Behavioral Adoption | 0.212 | 0.292 | *** | Significant |
| Perceived Risk | → | FinTech Behavioral Adoption | −0.077 | −0.082 | 0.208 | Not significant |

Notes: The indirect effect of perceived risks on FinTech adoption can be estimated by multiplying the coefficient of perceived risk (0.491) of the fear of COVID-19 and the coefficient of fear of COVID-19 (0.292) on FinTech behavioral adoption. Asterisks *** correspond to significance level at 1%.

For robustness, we applied the method used by [18] to conduct a resampling procedure called bootstrapping to confirm the results of the conventional mediation test. The procedure generates statistical significance of the path model coefficients in the model. The results of the bootstrapping procedure utilizing 1000 bootstrap samples, with bias-correction at 95%, are reported in Table 7. The results confirm that the direct effect of perceived risk on FinTech adoption is insignificant. Instead, the impact of perceived risk on FinTech behavioral adoption occurs indirectly through the fear of COVID-19. In other words, the fear of COVID-19 mediates the relationship between perceived risk and FinTech behavioral adoption.

**Table 7.** Direct vs. indirect effect tests using the bootstrapping procedure.

| Statistics | Indirect Effect | Direct Effect |
| --- | --- | --- |
| Bootstrapping Estimate | 0.143 | −0.082 |
| Bootstrapping *p*-value | 0.001 | 0.241 |
| Results | Significant | Not significant |

## 5. Discussion

The results from the direct effect models suggest that bank consumers' decisions to adopt FinTech services are strongly driven by their positive perceptions of the innovations. These findings are consistent with [11,12,17–20,46], supporting the importance of convenience [11,12,17,20], rapid and seamless transactions [17,20,40], and economic efficiency [11,12]. The results corroborate FinTech services providing a solution to the circumstances at the time when in-person activities were constrained due to the pandemic. This conjecture corroborates the evidence, which shows that fear of COVID-19 significantly strengthens the relationship between perceived benefits and FinTech services adoption. This finding supports the PMT [34] and explains the sudden increase in FinTech adoption following the pandemic [27,55]. More importantly, it also implies that fear of COVID-19 is a genuine concern for the public [25,55]. FinTech offers solutions to the affected consumers by enabling 24/7 worldwide access to contactless financial services, ensuring consumers' safety [25] from contagious physical contacts.

This study also demonstrates that perceived benefits have a more significant effect on FinTech behavioral adoption than perceived risks, consistent with previous studies [15–20]. The results support the NVF in suggesting that consumers adopt a technology if they perceive its benefits to outweigh its risks [33]. Meanwhile, the finding which indicates that bank consumers' perception of risk does not affect their FinTech use is similar to other studies in developing countries [18,19,37]. Some studies have associated the insignificant impact of perceived risks with the respondents' attributes. The younger generations (Gen Z or millennials) are tech-savvy and tech-dependent [40]. They would use FinTech services as a more efficient way of completing tasks without investing time and energy in physically carrying out the transactions. In addition, they have minimal exposure to cyber-attacks because they do not have many financial resources. Online, digital, and internet technologies also offer safety [25], a top priority for consumers during a health crisis. Other studies, such as [17], associated the indifferent attitude toward risks with the user's gender. Male respondents were suggested to care less about risk than their female counterparts.

The result of perceived risk is concerning. It corroborates [21], who asserted that the failure to anticipate and manage risks and uncertainties is the cause of bank consumers suffering most financial problems. It also implies that the consumers ignore FinTech risks, although cybercrime incidents are high. Cyber-attacks can incriminate consumers/users regardless of their financial resources. For instance, hackers can "virtually" make purchases and "virtually" charge it on a client's credit card who has inputted their data into the FinTech service provider space. Being indifferent toward FinTech risks can lead to serious financial troubles for FinTech users in the short term [11,12]. It could also lead to the destruction of FinTech business ecosystems in the long term [11,12]. Rationally, consumers will stop using and cut their loyalty to products or services that give them negative experiences [40].

Discovering that perceived risk is insignificant in his study, ref. [18] asserted that FinTech service providers should not neglect the risks of their services as the consumers would reassess their exposure to FinTech risks once the pandemic is over. Further examination is necessary because it indicates that consumers underestimate the potential threats of risks. Moreover, previous studies have shown that perceived risk significantly affects users' intention, adoption, or the continuance intention to use FinTech [11,12,15,19,36,39].

This study reveals the role of perceived risks by linking it with the fear of COVID-19. The results demonstrate that the pandemic has changed consumers' behavior in spending and purchasing [23,24] and their choices of payment and transfer methods. More specifically, different levels of COVID-19 fear among bank consumers result in different perceptions of how FinTech adoption benefits them. Perceived risk, which has no significant direct effect on FinTech behavioral adoption, has a significant indirect impact on FinTech behavioral adoption through the mediating fear of COVID-19. These results lend strong evidence for [18]'s conclusion that consumers' fear of COVID-19 infection obscures their concerns about FinTech risks. The findings corroborate our proposition that a fear of cash carrying COVID-19 [26,29] is more prominent in countries such as Malaysia, where cash is dominant. In other words, under a normal situation, these bank consumers would have perceived FinTech services as too risky to adopt. However, the pandemic has changed their priority. The perception is that COVID-19 infection would cause greater threats to their health and those in their circles; therefore, FinTech is accepted as an indispensable solution to minimize the potential of being infected. This finding is crucial in understanding the behavioral shift that enables the adoption of contactless alternatives in light of the recent pandemic.

Finally, this study reveals that FinTech behavioral adoption positively and significantly impacts sustainability ($p < 0.01$). The result affirms previous studies positing the implication of FinTech adoption on sustainability and achieving the Sustainable Development Goals (SDGs) [2,41,48,52]. Given the circumstances in which this survey was conducted, the results validate FinTech's capability to drive financial inclusion. It allows people affected by the pandemic to continue with their daily transactions, thus sparing them from depriving conditions. The pandemic has forced consumers to switch to FinTech services, and the statistics have proven that the jump in FinTech adoption is significant. This incidence helps accelerate the country's progress toward a cashless society; therefore, it is more important than ever that the consumers are given positive experiences. As argued by [40], positive experiences will motivate adopters to accept FinTech services as their new way of performing transactions. This finding also supports previous research recommendations about the role of ICT and its applications, including FinTech, in building resilience and ensuring sustainability during crises and beyond [6,7,49,53,59]. It extends and validates the findings of [21], suggesting that technology adoption could contribute to sustainability. Studies by [6,59] were carried out in Poland; similar findings are documented in this study. The consistent findings prove that the contribution of FinTech to sustainability is supported in Malaysia's context. Hence, the results have increased the generalizability of the theoretical perspective on the relationship between technology adoption and sustainability.

## 6. Conclusions

This study examined whether perceived benefit and risk have influenced Fintech adoption during the COVID-19 pandemic, and whether this adoption contributes to sustainability. In light of the severe health threats consumers face during the pandemic, this study also examined the role of fear of COVID-19 in FinTech adoption. The relationships, modeled based on the integration between NVF, SIS theory, and PMT, were empirically tested on the data of 400 randomly selected bank consumers with FinTech experience in Malaysia. The target respondents were users of traditional banks, the incumbents in the financial services industry disrupted by FinTech. The results of the SEM path analysis reveal that perceived benefits significantly influence FinTech behavioral adoption; however, in contrast to the well-established positions, the perceived risk does not. Consistent with

the advent of efficiency-optimizing I-4.0 technologies, bank consumers perceive FinTech adoption to significantly contribute to sustainability. The MCFA reveals that the fear of COVID-19 moderates the relationship between perceived benefits and FinTech behavioral adoption. Results of the path analysis and bootstrapping procedure confirm that the fear of COVID-19 fully mediates the effect of perceived risk on FinTech adoption. Overall, this study has demonstrated that FinTech benefits are important considerations in its adoption. However, consumer sentiment (specifically, fear of COVID-19) plays a vital role in strengthening the impact of these benefits and establishing risks in the FinTech adoption equation. These findings are crucial in creating FinTech natives to realize its effects on economic, environmental, and social sustainability.

The results of this study have important practical implications. FinTech service providers, including banks, should leverage those benefits to develop their FinTech services. At the same time, they must take necessary measures to address the increasing number of FinTech cyber security incidents to sustain their adoption. The results reveal that adoption during the pandemic has been driven by a fear of COVID-19. This finding implies that bank consumers will likely reassess their FinTech experience and abandon it over concerns about FinTech insecurities once the pandemic is over. FinTech service providers should be committed to investing in the security of their services, protecting their consumers' data, and providing swift assistance to resolve any security breaches. From the policymaker's perspective, the significant impact of FinTech adoption on sustainability suggests the importance of transforming into a cashless society and digital economy for developing economies. For Malaysia specifically, the results support the national plan to leverage I-4.0 digital technologies to responsibly achieve a high-income economy status. Governments must amplify their support and commitment to promoting mass FinTech adoption to realize financial inclusivity, community resilience, and sustainability during crises and beyond. To a great extent, COVID-19 has served some beneficiaries with silver linings in their efforts to accelerate FinTech adoption. FinTech industry players and policymakers should exploit the consumers' behavioral shifts during the pandemic and turn them into FinTech natives, consumers embracing FinTech as new ways of efficiently handling financial transactions beyond the crisis. Policymakers can use the results presented herein to strategize interventions and sustainable long-term policies to harness FinTech for solving sustainability challenges in the real economy.

This study also has significant academic implications. It demonstrates that antecedents of FinTech adoption, specifically, perceived benefits, are universal factors. In contrast, consistent with previous studies, consumers in developing economies often view perceived risks as less critical. Although there are some indications that this condition is attributed to the respondents' demographics, this study cannot attest to the conjecture, because we did not test the demographic effects. Some studies argue that younger generations with limited financial resources are more risk-tolerant and less exposed to the risks. Future studies should address this limitation or use a more representative sample. This study has also discovered the fear of COVID-19 to be an important factor in determining FinTech adoption. The finding is an essential contribution to the literature on technology adoption because it introduces consumer sentiment as one of the key determinants. Future studies on FinTech, e-commerce, online learning, and other digital technologies should consider the role of user sentiment and fear of COVID-19 in shaping their behavioral adoption and experience.

**Author Contributions:** Conceptualization, S.A.B. and R.A.-R.; methodology, S.A.B. and R.A.-R.; software, S.A.B. and Z.A.; validation, S.A.B., R.A.-R. and Z.A.; formal analysis, S.A.B. and Z.A.; investigation, S.A.B. and R.A.-R.; resources, S.A.B. and R.A.-R.; data curation, S.A.B.; writing—original draft preparation, S.A.B.; writing—review and editing, R.A.-R., S.A.B., Z.A. and A.A.; visualization, R.A.-R. and S.A.B.; supervision, R.A.-R. and A.A.; project administration, S.A.B. and R.A.-R.; funding acquisition, R.A.-R. and A.A. All authors have read and agreed to the published version of the manuscript.

**Funding:** This research was funded by the Ministry of Higher Education, Malaysia, under the Fundamental Research Grant Scheme, FRGS/1/2018/SS01/UKM/02/2. The views and opinions expressed in this article are those of the authors and do not necessarily reflect the views of the funding agency.

**Institutional Review Board Statement:** Not applicable.

**Informed Consent Statement:** Not applicable.

**Data Availability Statement:** The data presented in this study are available on request from the corresponding author.

**Acknowledgments:** The authors wish to acknowledge valuable feedback from Robert Faff, Chan Zhong Yang, and Suzana Mohamad Said.

**Conflicts of Interest:** The authors declare no conflict of interest. The funders had no role in the design of the study; in the collection, analyses, or interpretation of data; in the writing of the manuscript, or in the decision to publish the results.

## Appendix A

Figure A1 shows that the total transaction value of FinTech services worldwide was USD 12.106 trillion in 2021, with digital payments accounting for the largest proportion: 61%. The industry is expected to grow at a CAGR of 17% to reach USD 26.817 trillion by 2026.

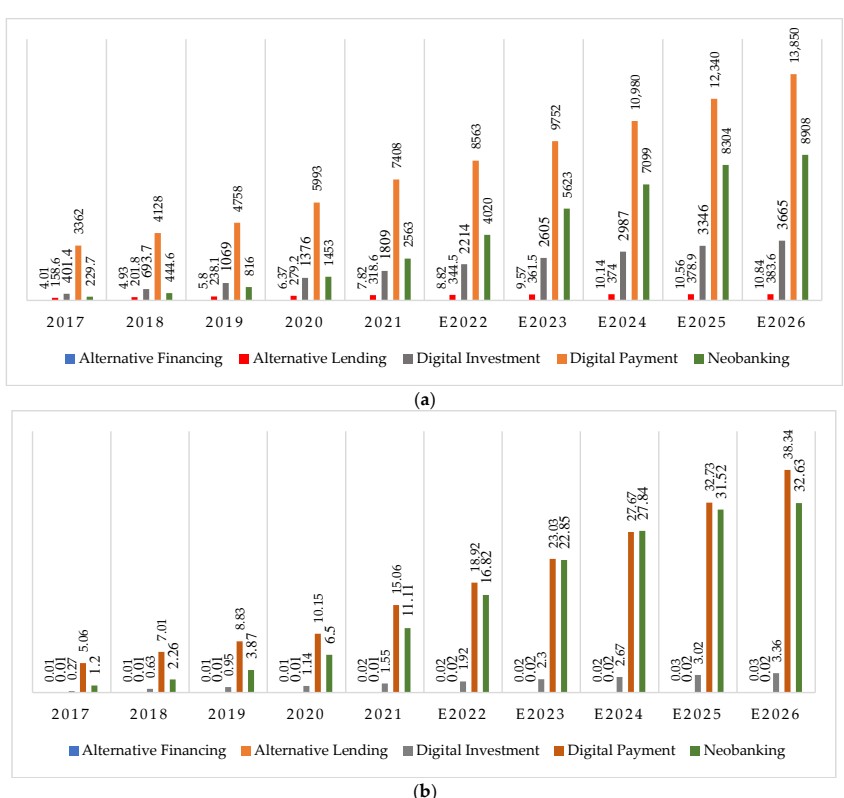

**Figure A1.** Transaction value of FinTech services in USD billions: (**a**) worldwide; (**b**) in Malaysia. Notes: Statista categorizes FinTech services into 5 segments: (1) alternative financing, which includes crowd investing and crowdfunding; (2) alternative lending, which includes crowdlending (business) and marketplace lending (consumers); (3) digital investment, which includes neobrokers and robo-advisors; (4) digital payments, which include mobile point-of-sale (POS) payments, digital remittances, and digital commerce; and (5) neobanking. Statista noted that the transaction value is calculated using current exchange rates. E affixed to years represents "estimated" values for 2022 onward. The data shown do not yet reflect the market impacts of the Russia–Ukraine war. Source of data: Statista.

**Table A1.** Construct, items, and sources.

| Constructs | Code | Sub-Constructs and Item Statements | Sources |
|---|---|---|---|
| | Non-monetary Benefits Dimension | | |
| Perceived Benefits Construct | NB1 | Using FinTech allows me to use financial services anytime | [11] |
| | NB2 | Using FinTech allows me to use financial services anywhere | [11] |
| | NB3 | Using FinTech makes it easier to access financial services | [19] |
| | NB4 | Using FinTech is easy for me (in setting up, configuring, and using the service) | [19] |
| | NB5 | Using FinTech is more convenient than traditional financial services | [30] |
| | NB6 | Using FinTech avoids much unnecessary hassle for me (e.g., going to the branches, avoiding traffic) | [16] |
| | NB7 | Using FinTech enables me to use financial services more quickly (contactless) | [19] |
| | NB8 | Using FinTech allows me to use financial services more effectively (save time and effort) | [19] |
| | NB9 | Using FinTech eliminates the time-consuming payment processes (the speed at checkout) | [16] |
| | Monetary Benefits Dimension | | |
| | MB1 | Using FinTech allows me to save money (e.g., discounts, promotions, coupons) | |
| | MB2 | Using FinTech allows me to use various financial services at a low cost | [11] |
| | MB3 | Using FinTech allows me to expect financial gains (e.g., cashback, higher interest, vouchers, rewards) | |
| | Personal Data Protection Risk Dimension | | |
| Perceived Risk Construct | PR1 | Using FinTech makes me worried about the abuse of my financial information (e.g., transaction information) | [11] |
| | PR3 | Using FinTech makes me worried that other people might steal my account information | [11] |
| | PR4 | I think the cybersecurity risk is much higher using FinTech than traditional financial services | [11] |
| | PR5 | The chances of using FinTech and losing control over my personal information privacy are high | [16] |
| | PR6 | Signing up and using FinTech would lead me to lose privacy because my personal information would be used without my knowledge | [16] |
| | PR7 | I am concerned that FinTech is collecting too much personal information from me | [16] |
| | PR8 | I am concerned that FinTech allows unauthorized persons to hack into my personal information | [45] |
| | PR9 | Using FinTech, I am concerned about the privacy of my personal information during a transaction | [45] |
| | PR12 | I worry about the way FinTech providers respond to financial losses or financial information leakages | [45] |

**Table A1.** *Cont.*

| Constructs | Code | Sub-Constructs and Item Statements | Sources |
|---|---|---|---|
| | Regulatory Risk Dimension | | |
| | RR1 | I am reluctant to use FinTech due to insufficient regulations | |
| | RR2 | I doubt FinTech due to its unclear legal status | [11] |
| | RR3 | I doubt FinTech due to a lack of regulations | |
| | RR4 | It is difficult to use various FinTech applications due to the lack of regulations | |
| | Financial Risk Dimension | | |
| | FR1 | Using FinTech, I am more likely to experience financial fraud or payment fraud | [11] |
| | FR2 | Using FinTech, I may suffer financial losses due to a lack of interoperability with other services | [11] |
| | FR3 | Using FinTech, I may suffer a monetary loss due to fluctuations in exchange rates | [45] |
| | FR4 | Using FinTech, I may suffer a monetary loss due to fluctuations in cryptocurrencies | [45] |
| | To make payment . . . | | |
| FinTech Behavioral Adoption * | FAB1 | I always use internet/online banking (using a browser, e.g., Maybank2u.com, CIMBclicks.com) | |
| | FAB2 | I always use a mobile banking app (using a smartphone, e.g., Maybank app, CIMB app) | [58,59] |
| | FAB3 | I always use contactless debit/credit/prepaid cards (e.g., PayWave, PayPass, ExpressPay) | |
| | FAB4 | I always use e-wallets (e.g., Touch'nGo, GrabPay, Boost, BigPay, WeChat Pay, AliPay, MAE) | |
| Fear of COVID-19 | FC1 | I am afraid of COVID-19 | |
| | FC2 | It makes me uncomfortable to think about COVID-19 | |
| | FC3 | My hands become sweaty when I think about COVID-19 | |
| | FC4 | I am afraid of losing my life because of COVID-19 | [27] |
| | FC5 | I feel uneasy when watching news and stories about COVID-19 on social media | |
| | FC6 | I worry about getting COVID-19 | |
| | FC7 | I am nervous when I think about getting COVID-19 | |
| Sustainability | S1 | FinTech adoption could reduce costs (e.g., through lower purchase prices of goods/services on the internet, eliminating travel expenses, and lower costs of communication over the internet than telephone or personal communication) | [6] |
| | S3 | FinTech adoption could expand existing knowledge and skills along with gaining new ones (e.g., including digital knowledge and skills, financial knowledge and awareness) | |

**Table A1.** *Cont.*

| Constructs | Code | Sub-Constructs and Item Statements | Sources |
|---|---|---|---|
| | S4 | FinTech adoption could increase the security of people and social groups through access to information and dissemination of information on various dangers and risks (e.g., alerts on phishing, scam calls, pandemic-related issues) | [6] |
| | S5 | FinTech adoption could reduce social exclusion due to age, education, place of residence, or disability, which causes difficult participation in banking and finance and limited or difficult access to financial services | |
| | S6 | FinTech adoption could reduce energy consumption (e.g., fuel) and increase protection of the environment through FinTech | |

\* Note: FinTech payment services via an online foreign exchange and cryptocurrency e-wallets were dropped from the model estimation after the pre-test.

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
