# Peer review of "Benefit–Risk Perceptions of FinTech Adoption for Sustainability from Bank Consumers’ Perspective: The Moderating Role of Fear of COVID-19"

_sustainability, doi:10.3390/su14148357_

Round 1

Reviewer 1 Report

Congratulations for your paper. It is challenging and indeed it fills a gap in the theory and practice of fintech development. However your paper can be improved. Here are my recommendations:

A) In the Introduction section make a small discussion about the importance of customer experience as a factor that contributed to the development of fintech sector. Suggested references: 

  1. Gomber, P.; Kauffman, R. J.; Parker, C.; Weber, B. W. On the Fintech revolution: Interpreting the forces of innovation, disruption and transformation in financial services. Journal of Management Information Systems 2018, 35, 220-265. https://doi.org/10.1080/07421222.2018.1440766
  2. Barbu, C.M.; Florea, D.L.; Dabija, D.C.; Barbu, M.C.R. Customer experience in fintech. J. Theor. Appl. Electron. Commer. Res. 2021,
    16, 1415–1433. https://www.mdpi.com/0718-1876/16/5/80
  3. Buckley, R.P.; Webster, S. Fintech in developing countries: Charting new customer journeys. Journal of Financial Transformation 2016, 44, 151–159.

 B) Please discuss in-depth your results, with an emphasize on theoretical and practical implications. You do so, whether by adding a Discussion section or by augmenting your Conclusions . As it is now, section 4. Results and Discussion is only about the results.  

C) Carefully review your English to improve readability.

Author Response

REVIEWER 1

Congratulations for your paper. It is challenging and indeed it fills a gap in the theory and practice of fintech development. However your paper can be improved. Here are my recommendations:

  1. A) In the Introduction section make a small discussion about the importance of customer experience as a factor that contributed to the development of fintech sector. Suggested references: 
  2. Gomber, P.; Kauffman, R. J.; Parker, C.; Weber, B. W. On the Fintech revolution: Interpreting the forces of innovation, disruption and transformation in financial services.  Manag. Inf. Syst.201835, 220-265. https://doi.org/10.1080/07421222.2018.1440766
  3. Barbu, C.M.; Florea, D.L.; Dabija, D.C.; Barbu, M.C.R. Customer experience in fintech. Theor. Appl. Electron. Commer. Res. 2021, 16, 1415–1433. https://doi.org/10.3390/jtaer16050080
  4. Buckley, R.P.; Webster, S. Fintech in developing countries: Charting new customer journeys. The Capco Institute Financ. Transform.2016, 44, 151–159. https://www.capco.com/Capco-Institute/Journal-44-Financial-Technology.

Authors’ Responses: Thank you for your suggestion. 1) I have included a brief mention of customer experience in the Introduction section (pg para, lines) and cited it in several other places in the paper. 2) I cited all suggested references in the revised paper; see Refs [19,25,27].

B) Please discuss in-depth your results, with an emphasis on theoretical and practical implications. You do so, whether by adding a Discussionsection or by augmenting your As it is now, section 4. Results and Discussion is only about the results.  

Authors’ Responses: Thank you. We separated the Discussion from the Results section, and elaborate on the results in the Discussion now.

C) Carefully review your English to improve readability.

Authors’ Responses: Thank you. We have gone through the revised manuscript to make sure the readability is improved.

Reviewer 2 Report

The basic idea of ​​the research is very good. After all, with the advancement of digitalisation, financial services must be developed, in which environmental challenges must be met as various environmental impacts that can affect sustainability.

Author Response

REVIEWER 2

The basic idea of â€‹â€‹the research is very good. After all, with the advancement of digitalisation, financial services must be developed, in which environmental challenges must be met as various environmental impacts that can affect sustainability.

Authors’ Responses: Thank you. We appreciate your positive feedback.

Reviewer 3 Report

Dear authors,

I appreciate having the opportunity to review the manuscript entitled “Benefit-Risk Perceptions of Fintech Adoption for Sustainability from Bank Consumers’ Perspective: The Moderating Role of Fear of COVID-19” (sustainability-1736662).

The subject of this research is a very interesting one, being topical and representative of the behavioral changes that the coronavirus pandemic has brought among consumers. The paper is well structured, and the hypotheses are supported by the analysis of the literature. The results were obtained from CFA and SEM analyzes, but also from various applied statistical tests.

Although the authors have made considerable efforts to develop this paper, I believe that the current version of the manuscript could be slightly improved. I want to provide some suggestions for the improvement of this paper as follows:

Line 154- You are missing a square bracket for the references mentioned ( “….and economic efficiency 2,6-8,11,26].

Figure 1 - It would be good to redo the left side of the model, more precisely the part related to First-order Construct. These items should be seen more clearly in the boxes around them, and the arrows that go from them to Second-order Construct should look better.

In Tables 4 and 6, what does *** mean?

In Table 4, at Prob .676, you forgot to add a 0 in front of the comma.

Best of luck!

Author Response

REVIEWER 3

I appreciate having the opportunity to review the manuscript entitled “Benefit-Risk Perceptions of Fintech Adoption for Sustainability from Bank Consumers’ Perspective: The Moderating Role of Fear of COVID-19” (sustainability-1736662).

  1. The subject of this research is a very interesting one, being topical and representative of the behavioral changes that the coronavirus pandemic has brought among consumers. The paper is well structured, and the hypotheses are supported by the analysis of the literature. The results were obtained from CFA and SEM analyzes, but also from various applied statistical tests.
    Authors’ Responses: Thank you for taking an interest in our paper.
  2. Although the authors have made considerable efforts to develop this paper, I believe that the current version of the manuscript could be slightly improved. I want to provide some suggestions for the improvement of this paper as follows:
  3. Line 154- You are missing a square bracket for the references mentioned ( “….and economic efficiency 2,6-8,11,26]. - done
  4. Figure 1 - It would be good to redo the left side of the model, more precisely the part related to First-order Construct. These items should be seen more clearly in the boxes around them, and the arrows that go from them to Second-order Construct should look better. - Figure 1 has been improved.
  • In Tables 4 and 6, what does *** mean? - definition added
  1. In Table 4, at Prob .676, you forgot to add a 0 in front of the comma. - 0 added
  2. Best of luck! - thank you professor

Authors’ Responses: Thank you. We have incorporated all your suggestions, including improving Figure 1.

Reviewer 4 Report

I think the article raises an interesting and topical issue. The research has been prepared and conducted in a correct way and the results obtained are of practical use. However, in my opinion, the authors should carry out a more thorough analysis of the obtained results with the current state of knowledge. Moreover, I think that methodological issues should be clarified. 

Author Response

REVIEWER 4

  1. I think the article raises an interesting and topical issue. The research has been prepared and conducted in a correct way and the results obtained are of practical use.

Authors’ Responses: Thank you for your positive feedback.

  1. However, in my opinion, the authors should carry out a more thorough analysis of the obtained results with the current state of knowledge.

Authors’ Responses: Thank you. We have elaborated our discussion on the results (by offering explanations and compare with the results of previous studies) in a new section, Discussion.

  1. Moreover, I think that methodological issues should be clarified

Authors’ Responses: Thank you for your positive feedback.

Reviewer 5 Report

Abstract
The authors need to explain the sampling methods for data collection and research methods for data analysis. Report study limitations and future recommendations.

Introduction
Briefly add research objectives in the last paragraph of the abstract ending lines. In the introduction, the study uses up-to-date references to identify issues and research gaps. This study provides sufficient theoretical support. However, how this research contributes to the entire field of research needs to be addressed. The study must discuss who the potential beneficiaries of this research are? -Especially, the introduction section needs to re-organize. The major debate or Argument is not clear stated in the introduction session. Hence, the contribution debates are weak in this manuscript. I would suggest that the author enhance your literature discussion and arrive your debate or Argument. I suggest authors to revise the introduction part.

In the introduction, you need to connect the state of the art to your paper goals. Please

follow the literature review by a clear and concise state of the art analysis. This should

clearly show the knowledge gaps identified and link them to your paper goals. Please

reason both the novelty and the relevance of your paper goals. Clearly discuss what the

previous studies that you are referring to.What are the Research Gaps/Contributions? Please note that the paper may not be considered further without a clear research gap and novelty of the study.

The literature
I suggest the authors to add more literature to make the literature strong. In this study, literature part, the conceptualization of each study variable can be added. Additionally, explain the relationship between the variable of the study. Please explains the use of theory to develop this model. Literature Review has the chance to be further improved: it seems that the authors have made the retrospection. However, via the review, what issues should be addressed? What is the current specific knowledge gap? What implication can be referred to? The above questions should be answered. Authors need to propose their study.In the introduction, clearly state´what is sustainability. See the following.

https://doi.org/10.3390/app12031521

https://doi.org/10.1007/s11356-022-18689-y

Methods and Results
The methods and results show good explanation. I suggest refining these sections to remove minor errors. Add justification of this study sample. Explain which method was followed for the required sample size calculation.

Discussion
Please make sure your conclusions' section underscore the scientific value added of your paper, and/or the applicability of your findings/results, as indicated previously. Please revise your conclusion part into more details. Basically, you should enhance your contributions, limitations, underscore the scientific value added of your paper, and/or the applicability of your findings/results and future study in this session. The discussion is relatively simple and insufficient. I recommend strengthening the comparison with previous research. Please compare the results in this study with those in previous studies.

Discuss the study findings here. The discussion and conclusion are appropriately written and require no changes. The manuscript does not answer the following concerns: Why is it timeliness

to explore such a study? What makes this study different from the previously published studies? Are there any similarly findings in line with the previously published studies? Are the findings different from prior academic studies that were conducted elsewhere, if any?

Implications:

Please explain your results into steps and links to your proposed method.

I would like to request the author to emphasize the contributions practically and academically in the implication session.

Concluding remarks

Remove minor grammar errors. Refine the conclusion and conclude this section with key findings. Follow the comments and improve this study. Overall, the paper qualifies for publication in the journal with the above-mentioned minor corrections.

Author Response

REVIEWER 5

Authors’ Responses: Thank you for your positive feedback. We addressed your comments and suggestions, as described point-by-point below:

A) Abstract

  1. The authors need to explain the sampling methods for data collection and research methods for data analysis. Report study limitations and future recommendations.
  2. Briefly add research objectives in the last paragraph of the abstract ending lines.

Authors’ Responses: Thank you for the suggestions. We have restructured the abstract. While making sure it does not go over the words limit(250), the abstract now describes;  

  • data collection (research method and SEM). The limitations and future recommendations are retained in the Conclusion section (last paragraph).
  • objectives, in the first few sentences at the beginning of abstract

B) Introduction

  1. In the introduction, the study uses up-to-date references to identify issues and research gaps. This study provides sufficient theoretical support. However, how this research contributes to the entire field of research needs to be addressed.

Authors’ Responses: This study has three contributions based on issues in the literature, industry, and ecosystem (COVID-19). These contributions were explained in the 2nd last paragraph of the Introduction section (pg 4, lines 154 to 164).

  1. The study must discuss who the potential beneficiaries of this research are?

Authors’ Responses: We added the beneficiaries of this study to the Introduction section (page 3, last paragraph, lines 164 – 169. Pg.4).

  1. Especially, the introduction section needs to re-organize.

Authors’ Responses: We have reorganized the Introduction section, and added/changed it according to the following comments, while also trying our best to make sure a smooth presentation of the discussion and arguments.

  1. The major debate or Argument is not clear stated in the introduction session. Hence, the contribution debates are weak in this manuscript. I would suggest that the author enhance your literature discussion and arrive your debate or Argument. I suggest authors to revise the introduction part.

Authors’ Responses: The arguments of this study are as follows:

  • Insufficient attention of FinTech risks: As suggested by [1-3], previous studies on FinTech focus on its benefits, while insufficient attention is paid to risks to address the increasing of cybercrimes incidents resulting from FinTech usage (see the beginning of the Introduction section, pg. 1-2). In the same paragraph, we proposed to close the literature gap by considering both the perceived benefits and risks of FinTech services in Net Valence Framework. In para 4, pg 2 to para 2 pg 3), again we stressed on the burgeoning dangers and risks of FinTech.
  • Impact of FinTech on Sustainability: In the following paragraph (para 3, pg 3), we discussed the potential contribution of FinTech adoption to sustainability, through financial inclusion (access to underbanked and unbanked etc).
  • Impact of COVID-19 on FinTech adoption: In the following paragraph (para 4, pg 3), we explained how FinTech adoption has increased tremendously as a result of the government’s physical and social containments. This argument is based on the policy responses to consumers’ fear of infection through cash and coins. Thus, this study examines if fear of COVID-19 could explain the behavioral shift toward FinTech adoption.

  1. In the introduction, you need to connect the state of the art to your paper goals. Please follow the literature review by a clear and concise state of the art analysis. This should clearly show the knowledge gaps identified and link them to your paper goals. Please reason both the novelty and the relevance of your paper goals. Clearly discuss what the previous studies that you are referring to. What are the Research Gaps/Contributions? Please note that the paper may not be considered further without a clear research gap and novelty of the study.

Authors’ responses: Thank you. As stated in (3), we tried our best to highlight the gaps in the literature and address them in the contributions of our study.

C) The literature

  1. I suggest the authors to add more literature to make the literature strong. In this study, literature part, the conceptualization of each study variable can be added.

The conceptualization of te main variables is provided in the Litertuare Reviewe section. Please refer to the following checklist to check where we provided the conceptualization of the main variables:

  1. FinTech services – line 177, pg 4.
  2. Perceived benefits – line 185, pg 4 & 233, pg 5, with specific examples mention in many places.
  • Perceived risks – line 190, pg 4 & 233, pg 5
  1. Fear of COVID – 363 – 366, pg. 8
  2. Sustainability – lines 287-292, pg 6.
  3. Additionally, explain the relationship between the variable of the study.

We have presented the Lit. Review by sections to focus on the relationship between the variables. Specifically;

Section 2.1: NVF to explain the relationship between perceived benefits and risks and FinTech adoption.

Section 2.2: SIS argument to hypothesize the relationship between FinTech adoption and sustainability

Section 2.3: PMT argument to hypothesize the relationship between FinTech adoption and fear of COVID-19

  1. Please explains the use of theory to develop this model. Literature Review has the chance to be further improved: it seems that the authors have made the retrospection. However, via the review, what issues should be addressed?

There are three main theories that we use as arguments for the hypothesized relationships in this study:

  1. Net Valence Framework (NVF) – to argue that both perceived benefits and risks must be considered in the adoption decision,
  2. Sustainable Information Society (SIS) – to argue that the adoption of Information-based technologies (ICT) like FinTech will contribute to social, economic, and environmental sustainability, and
  3. Protection Motivation Theory – to argue that consumers are motivated to protect their well-being from the fear of being infected by COVID-19 in their decision to adopt FinTech.

We discussed the theories throughout the paper but explained them in the Literature Review, specifically sections 2.1 – 2.3 as described in Item (C-2) above.

  1. What is the current specific knowledge gap? What implication can be referred to? The above questions should be answered. Authors need to propose their study. In the introduction, clearly state what is sustainability. See the following.

https://doi.org/10.3390/app12031521

https://doi.org/10.1007/s11356-022-18689-y

Authors’ Responses:

  1. We explained in the Introduction and Lit. Review sections the following knowledge gaps are as follows: i) lack of empirical attention on perceived risks despite the increasing cybercrime incidents, ii) scant evidence on the link between FinTech services and sustainability, and iii) perhaps no studies have examined the impact of user sentiment, in particular, fear of COVID-19 on FinTech behavioral adoption.
  2. The academic and practical implications of this study are provided in the Conclusion section, and briefly in the Introuction section.
  3. Sustainability is explained in length in the Literature Review section (Section 2.2) because the Introduction section is rather too long.

We have referred to the two suggested studies (shown below) and cited the second one [see Ref. 27]. The first study is a very well-written paper on leveraging circular economy to achieve sustainability but appears to be outside of the scope of our study:

- Awan, U.; Sroufe, R. Sustainability in the Circular Economy: Insights and Dynamics of Designing Circular Business Models.  Sci.202212, 1521. https://doi.org/10.3390/app12031521

- Khan, H.u.R.; Usman, B.; Zaman, K.; Nassani, A.A.; Haffar, M. Muneer, G. The Impact of Carbon Pricing, Climate Financing, and Financial Literacy on COVID-19 Cases: Go-For-Green Healthcare Policies. Environ Sci Pollut Res2022, 2935884–35896 (2022). https://doi.org/10.1007/s11356-022-18689-y

D) Methods and Results

  1. The methods and results show good explanation.

Authors’ Responses: Thank you for the positive feedback.

  1. I suggest refining these sections to remove minor errors.

Authors’ Responses: Thank you. We have improved these sections according to suggestions by all reviewers.

  1. Add justification of this study sample.

Authors’ Responses: Thank you. We have provided the justification for the sample in Section 3.3 and in Section 6. Conclusion (paragraph 1)

  1. Explain which method was followed for the required sample size calculation.

 Authors’ Responses: This study uses the simple random sampling method, through which 400 questionnaires are randomly selected from 1,279 usable ones. The decision to cap at 400 is to suit the SEM method. The sample frame is developed by administering the questionnaires through online platforms using purposive and snowball sampling methods. These descriptions are provided in Section 3.3.

E) Discussion

  1. Please make sure your conclusions' section underscore the scientific value added of your paper, and/or the applicability of your findings/results, as indicated previously.

Authors’ responses: This comment is addressed as explained in Part G later.

  1. Please revise your conclusion part into more details. Basically, you should enhance your contributions, limitations, underscore the scientific value added of your paper, and/or the applicability of your findings/results and future study in this session. 

Authors’ responses: This comment is addressed as explained in Part G later.

  1. The discussion is relatively simple and insufficient. I recommend strengthening the comparison with previous research. Please compare the results in this study with those in previous studies.

Authors’ Responses: Thank you. We have elaborated our discussion on the results (by offering explanations and compare with the results of previous studies) in a new section, Discussion.

  1. Discuss the study findings here. The discussion and conclusion are appropriately written and require no changes.

Authors’ responses: Dear editor, there is a mixed comment here. On one hand, the reviewer stated that the discussion and conclusion are appropriately written and require no changes (see statement highlighted in blue above). We still give our best to make sure our paper addresses the reviewer’s remaining questions in this section;

  1. The manuscript does not answer the following concerns: Why is it timeliness to explore such a study?

Authors’ responses: The explanation is provided in the Introduction – lines 13-14, pg. 2.

‘]. In light of the tremendous FinTech financial service uptakes due to COVID-19 and increasing cybercrime incidents, this study is a timely attempt to close the gaps in the literature by considering the antecedents of FinTech adoption from both benefit and risk perspectives [22], specifically in the Net Valence Framework [23].’

  1. What makes this study different from the previously published studies?

Authors’ responses: This question is answered by describing the contributions of this study (lines 154-164, pg 4).

  1. Are there any similarly findings in line with the previously published studies?

 Authors’ responses: Certainly, and the Discussion section has addressed this question.

  1. Are the findings different from prior academic studies that were conducted elsewhere, if any?

Authors’ responses: Certainly, and the Discussion section has addressed this question.

F) Implications

  1. Please explain your results into steps and links to your proposed method.

Authors’ responses: We address these comments by 1) dedicating section 4. Results to report the results, according to the sequence of our hypotheses, and 2) creating a separate section 5. Discussion, to discuss possible explanations for the results and compare them with results of previous studies, in that sequence, one after another.

  1. I would like to request the author to emphasize the contributions practically and academically in the implication session.

Authors’ responses: We have reorganized and improved our Conclusion section. Paragraphs 2 to 4 in the section are dedicated to explaining the research (academic) and practical (management and policy) contributions.

G) Concluding remarks

  1. Remove minor grammar errors. Refine the conclusion and conclude this section with key findings.

Authors’ responses: We have refined the paper and proofread it as suggested.

  1. Follow the comments and improve this study.

Authors’ responses: We have addressed all comments.

  1. Overall, the paper qualifies for publication in the journal with the above-mentioned minor corrections.

Authors’ responses: Thank you, Professor. As explained in our responses above, we have addressed all comments to the best of our ability.

Round 2

Reviewer 5 Report

Dear authors, thanks for having adopted some of my comments. However, the paper still suffers of several issues.

The authors have responded to all editors' comments in a suitable manner. As a result, the manuscript has improved enormously. Nevertheless, some further work is needed in order to make the text easier to follow by the readers.
Please follow the journal author's instructions. It would be useful for the reader to follow it. In general, the paper needs better organization.

The abstract should state briefly the purpose of the Research, the principal results and major conclusions. An abstract is often presented separately from the article, so it must be able to stand alone. Please underscore the scientific value added of your paper in your abstract and Introduction.

The major defect of this study is the debate or Argument is not clear stated in the introduction session. Hence, the contribution is weak in this manuscript. I would suggest that the author enhance your theoretical discussion and arrive your debate or Argument.

I have  serious concern on introduction section.Especially, the introduction section needs to re-organize. The major debate or Argument is not clear stated in the introduction session. Hence, the contribution debates are weak in this manuscript. I would suggest that the author enhance your literature discussion and arrive your debate or Argument.

Methods section determines the results. Kindly focus on three basic elements of the methods section.
a. How the study was designed?
b. How was the study carried out?
c. How were the data analyzed?

Please explain your results into steps and links to your proposed method.
I would like to request the author to emphasize the contributions practically and academically in the implication session.
Please make sure your conclusions' section underscore the scientific value added of your paper, and/or the applicability of your findings/results, as indicated previously. Please revise your conclusion part into more details. Basically, you should enhance your contributions, limitations, underscore the scientific value added of your paper, and/or the applicability of your findings/results and future study in this session.

Author Response

Dear Professor,

First of all, as suggested in the checked box: Extensive editing of English language and style required

[Journal Editorial Team: If one of the referees has suggested that your manuscript should undergo extensive English revisions, please address this issue during revision.]

Authors’ Response: We had this paper edited by the MDPI’s Editing Service. The changes suggested by the MDPI’s team have been incorporated in this revised version. The original edited paper (ref: english-edited-46453) is attached for your reference.

Reviewer’s Comments and Suggestions for Authors:

Dear authors, thanks for having adopted some of my comments. However, the paper still suffers of several issues.

  1. The authors have responded to all editors' comments in a suitable manner. As a result, the manuscript has improved enormously. Nevertheless, some further work is needed in order to make the text easier to follow by the readers.

Authors’ Response: Thank you, Professor. We value your comments and suggestions as they help improve our paper. In this revision, we have given our best to address all of your concerns.

  1. Please follow the journal author's instructions. It would be useful for the reader to follow it. In general, the paper needs better organization.

Authors’ Response: We have rechecked the Journal Author’s Instruction and followed the guidelines to the best of our abilities.

  1. The abstract should state briefly the purpose of the Research, the principal results and major conclusions. An abstract is often presented separately from the article, so it must be able to stand alone.

Authors’ Response: Following the Journal’s Author Instruction, the Abstract must be limited to 200 words and contain the following components (we present the revised abstract here, by component, to ease your review):

  1. Background (broad research question and purposes):

Industry 4.0 technologies, designed to optimize efficiencies, are indisputable change agents for sustainability. In the context of financial technology (FinTech), the burgeoning question is how to create FinTech natives from the COVID-19 pandemic-induced adoption and realize FinTech’s impact on sustainability? Thus, this study carries these purposes: 1) to examine whether perceived benefits and risks affect FinTech services adoption; 2) to test the role of fear of COVID-19 in FinTech adoption; 3) to investigate whether FinTech adoption contributes to sustainability.

  1. Method (method or treatment applied):

The hypotheses derived from Net Valence Framework, Sustainable Information Society theory, and Protection Motivation Theory, are tested using the Structural Equation Modeling (SEM). Our online survey of bank consumers in Malaysia between December 2021 and February 2022 yields 1,279 usable questionnaires, randomly selected to generate 400 respondents.

  1. Results (summarize the article’s findings):

The results reveal that: 1) perceived benefits significantly influence FinTech adoption while perceived risk does not; 2) fear of COVID-19 moderates the perceived benefits-FinTech adoption relationship and fully mediates the perceived risk-FinTech adoption relationship; 3) FinTech adoption significantly affects sustainability.

  1. Conclusion (main conclusions and interpretation):

This study demonstrates that FinTech adoption models must exploit consumer sentiment (e.g., fear) to optimize FinTech’s benefits and risks, thereby creating FinTech natives to realize its impacts on economic, environmental, and social sustainability.

Abstract word counts = 200

  1. Please underscore the scientific value added of your paper in your abstract and Introduction.

Authors’ Response: The scientific value added of our paper is establishing consumer sentiment (in this case, fear of COVID-19) as an important factor to drive and sustain FinTech adoption, thereby creating FinTech natives to realize FinTech impact on sustainability.

In abstract: Lines 15 – 18.

In Introduction: Paragraph 1 in this section.

  1. The major defect of this study is the debate or Argument is not clear stated in the introduction session. Hence, the contribution is weak in this manuscript. I would suggest that the author enhance your theoretical discussion and arrive your debate or Argument.

Authors’ Response: We have improved our Introduction, clearly stating our Argument in the first 2 paragrapgh of the section, and strengthen our Contributions (last paragraph on page 3 to first paragraph on page 4).

  1. I have serious concern on introduction section. Especially, the introduction section needs to re-organize. The major debate or Argument is not clear stated in the introduction session. Hence, the contribution debates are weak in this manuscript. I would suggest that the author enhance your literature discussion and arrive your debate or Argument.

Authors’ Response: We have reorganized our Introduction. Note that other points are already explained in item 5 above.

The Journal’s Author Instruction suggests including hypothesis in the Introduction section. However, since our paper includes a Literature Review section, we develop our hypotheses in the LR section following discussion of theories.

  1. Methods section determines the results. Kindly focus on three basic elements of the methods section.
    How the study was designed?

Authors’ Response: Section 3.1 has been improved. It explains the how the instrument is adapted from several major studies, then it undergoes expert validation and pilot study before it is used in the survey (field study).

  1. How was the study carried out?

Authors’ Response: Section 3.3 is now improved to explain how the survey is conducted, how are the questionnaires distributed, to who it is administered, the sampling method, and the final sample.

  1. How were the data analyzed?

Authors’ Response: A new sub-section “3.2 Data Analytical Methods” is dedicated to present our explanation clearly. Briefly, the data are analyzed in 4 steps:

  1. CFA on field study data of 400 respondents (randomly selected from 1279 usable questionnaires) to determine validity
  2. Structural Equation Modeling (SEM) for the path analysis – direct and mediating effects
  3. Multi-group CFA – moderating effect
  4. We also use bootstrapingprocudure for robustness

  1. Please explain your results into steps and links to your proposed method.

The results are reported according to the following steps:

4.1 Profiles of respondents – standard in scholarly articles.

4.2 Results of CFA procedures validating the instrument – constructs and dimensions.

 4.3 Results of SEM Path Analysis – direct effects

4.4 Results of multi-group CFA (MCFA) on the moderating effect

4.5 Results of SEM Path Analysis – with the mediating effect, supported by the bootstrapping procedure as robustness check.

  1. On the Conclusion and Implication
    1. I would like to request the author to emphasize the contributions practically and academically in the implication session.

Authors’ response: In this paper, implications are presented in the Conclusion section. This section contains 3 paragraph; the first is the conclusion. The second discusses the practical implications:

The results of this study have important practical implications. FinTech services providers, including the banks, should leverage those benefits to develop their FinTech financial services. At the same time, they must take necessary measures to address the increasing FinTech cyber security incidents to sustain their adoption. The results reveal that adoption during the pandemic is driven by fear of COVID-19. It implies that bank consumers will likely reassess their FinTech experience and abandon it over concerns about FinTech insecurities once the pandemic is over. The FinTech service providers should be committed to investing in the security of their services, protecting their consumers’ data, and providing swift assistance to any security breaches. From the policymaker’s perspective, the significant impact of FinTech adoption on sustainability suggests the importance of transforming into a cashless society and digital economy for developing economies. For Malaysia specifically, the results support their plan to leverage I-4.0 digital technologies to achieve a high-income economy status responsibly. Governments must amplify their support and commitment to promoting mass FinTech adoption to realize financial inclusivity, community resilience, and sustainability during crises and beyond. To a great extent, COVID-19 has served some beneficiaries with silver linings in their efforts to accelerate FinTech adoption. FinTech industry players and policymakers should exploit the consumers’ behavioral shifts during this pandemic and turn them into FinTech natives, consumers embracing FinTech as new ways of efficiently handling financial transactions beyond the crisis. Policymakers can use the results in strategizing interventions and sustainable long-term policies to harness FinTech for solving sustainability challenges in the real economy.

The third paragraph explains the academic implications:

This study also has significant academic implications. It demonstrates that antecedents of FinTech adoption, specifically perceived benefits, are universal factors. On the contrary, consistent with previous studies, consumers in developing economies often view perceived risks as less critical. Although there are some indications that this condition is attributed to the respondents’ demography, this study cannot attest to the conjecture since it does not test the demographic effect. Some studies argue that the younger generation with limited financial resources is more risk-tolerant and less exposed to the risks. Future studies should address this limitation or use a more representative sample. This study has also discovered fear of COVID-19 as an important factor in determining FinTech adoption. The finding is essential to the literature on technology adoption because it introduces consumer sentiment as one of the determinants of technology adoption. Future studies on FinTech, e-commerce, online learning, and other digital technologies should consider the role of user sentiment and fear of COVID-19 in shaping their behavioral adoption and experience.

  1. Please make sure your conclusions' section underscore the scientific value added of your paper, and/or the applicability of your findings/results, as indicated previously.

Authors’ Response: Yes, it is as explained in the Conclusion section.

  1. Please revise your conclusion part into more details.

 Authors’ Response: Yes, the Conclusion section has been revised.

  1. Basically, you should enhance your contributions, limitations, underscore the scientific value added of your paper, and/or the applicability of your findings/results and future study in this session.

Authors’ Response: Yes, we have considered your suggestions.

INSTRUCTION FROM EDITORIAL TEAM – via email

  • Please check that all references are relevant to the contents of the

Dear editors,

We tried our best to reduce some references that were only cited initially to support the methodology issues. Now we have removed some references.

However, we’d like to request to reinstate the following reference which was dropped earlier due to the editor’s suggestion.

Ahorsu, D.K.; Lin, C.Y.; Imani, V.; Saffari, M.; Griffiths, M.D.; Pakpour, A.H. The Fear of COVID-19 Scale: Development and Initial Validation. Int. J. Ment. Health Addiction 2022, 20, 1537–1545. https://doi.org/10.1007/s11469-020-00270-8 . Please note that the article was cited as a 2020 publication (as in many papers) because it was published only in 27 March 2020).

We just realized that our fear of COVID-19 measurements are adapted from this source. We believe the editor could this article another consideration since it has made a significant impact on the literature. It has more 2800 citations and cited by articles published in high-impact journals. The followings are among articles citing this paper:

  • Cited on Pg 1, para 2: Taylor et al. 2020. (cited by 758) Development and Initial Validation of the COVID Stress Scale, Journal of Anxiety Disorder, 72, 102232. https://doi.org/10.1016/j.janxdis.2020.102232 (Elsevier)
  • Cited on Pg 3, 2nd last line, first column: Mertens, et al. (2020) (cited by 826). Fear of the coronavirus (COVID-19): Predictors in an online study conducted in March 2020, Journal of Anxiety Disorders, 74, https://doi.org/10.1016/j.janxdis.2020.102258
  • Cited in the Introduction section, line 2: Millroth & Frey (2021) (cited by 7). Fear and anxiety in the face of COVID-19: Negative dispositions towards risk and uncertainty as vulnerability factors, Journal of Anxiety Disorders, 83, 2021, 102454, https://doi.org/10.1016/j.janxdis.2021.102454.
  • Adapted by (pg 5, para 2, line 1): Saricali, M., Satici, S.A., Satici, B. et al.(2020) (cited by 78) Fear of COVID-19, Mindfulness, Humor, and Hopelessness: A Multiple Mediation Analysis. Int J Ment Health Addiction. https://doi.org/10.1007/s11469-020-00419-5 (Springer)
  • Adapter by (Ref. 55): Pilch I, Wardawy P, Probierz E (2021) (cited by 3) The predictors of adaptive and maladaptive coping behavior during the COVID-19 pandemic: The Protection Motivation Theory and the Big Five personality traits. PLoS ONE 16(10): e0258606. https://doi.org/10.1371/journal.pone.0258606
  • Adapted by (pg 4, column 1, 1st para). Alessandro et al. (2020) (cited by 85). The Anxiety-Buffer Hypothesis in the Time of COVID-19: When Self-Esteem Protects From the Impact of Loneliness and Fear on Anxiety and Depression, Frontiers in Psychology, 11, 2020, DOI=10.3389/fpsyg.2020.02177

  • Any revisions to the manuscript should be marked up using the “Track
    Changes” function if you are using MS Word/LaTeX, such that any changes can
    be easily viewed by the editors and reviewers.

Followed.

  • Please provide a cover letter to explain, point by point, the details
    of the revisions to the manuscript and your responses to the referees’

Provided.

  • If you found it impossible to address certain comments in the review
    reports, please include an explanation in your rebuttal.

Ok

  • The revised version will be sent to the editors and reviewers.

Noted.

Thank you again for the constructive comments and looking forward to receiving good news from your editorial team, soon.

Assoc. Prof. Dr. Ruzita Abdul-Rahim

UKM - National University of Malaysia

Round 3

Reviewer 5 Report

Introduction.

I greatly welcomed the improvements made by the authors in their introduction. However, my point is that this revised version of the manuscript did not succeed in offering an introduction that clearly states the study's research domain and emphasizes the relevance and originality of this contribution to the extant scientific literature. As I state in my previous revision letter, I see at least three major shortcomings in this introduction. I see that authors unable to build a strong research gap. This greatly undermines the quality and the relevance of this research. Secondly,  I am still unable to clearly see this scientific contribution's expected value. The authors will be able to enhance the attractiveness of their research.

- Please

reason both the novelty and the relevance of your paper goals. Clearly discuss what the

previous studies that you are referring to.What are the Research Gaps/Contributions? Please note that the paper may not be considered further without a clear research gap and novelty of the study.

Literature review and theoretical section

Unfortunately, I could not see relevant and compelling improvements in the theoretical framework, which still presents the major shortcomings that affected the original submission of this manuscript. In my own perspective, the authors are unable to build a fully-fledged and effective conceptual background in light of the references to the scientific literature they make. Rather, they seem to cherry-pick insights from previous scientific contributions and use them to work out several assumptions about the implications of big data governance and big data management on organizational innovativeness.

- I miss the conceptual nuances that allow us to understand how these phenomena are related. Of course, these nuances should be framed in lights of the theoretical background on which this paper is established. I also recommend the authors to adopt a more scientific style in arguing their research hypotheses. I rooted in the extant scientific debate, otherwise the study may be argued to suffer from limited conceptual reliability and consistency.

METHODS

The methodology now includes some additional insights, which may help the reader assess this study's dependability. However, these revisions are insufficient to support the study's consistency and dependability. The authors are still unable to argue the motivations that triggered them to focus on the  particularly industry.

Author Response

AUTHORS’ RESPONSES TO THE REVIEWER COMMENTS:

REVIEWERS 1 – 4: Satisfied after the 1st revision

REVIEWER 5:

As suggested in the checked box: Extensive editing of English language and style required

Authors’ Response: With all due respect, the revised version was already edited by the MDPI English Editing Service. Proof of editing can be requested from the Editorial Team.

Reviewer’s Comments and Suggestions for Authors

  1.  I greatly welcomed the improvements made by the authors in their introduction. However, my point is that this revised version of the manuscript did not succeed in offering an introduction that clearly states the study's research domain and emphasizes the relevance and originality of this contribution to the extant scientific literature. As I state in my previous revision letter, I see at least three major shortcomings in this introduction.
    1. I see that authors unable to build a strong research gap. This greatly undermines the quality and the relevance of this research.
    2. Secondly,  I am still unable to clearly see this scientific contribution's expected value. The authors will be able to enhance the attractiveness of their research.

- Please reason both the novelty and the relevance of your paper goals. Clearly discuss what the previous studies that you are referring to. What are the Research Gaps/Contributions? Please note that the paper may not be considered further without a clear research gap and novelty of the study.

Authors Response: Thank you sir, for your comments. However, we believe that all of your comments have been clearly addressed in our 2nd revision. Please go through the Introduction again.

The first three paragraphs built the arguments from theories and previous studies, followed by the motivation, objectives and contributions.

Industry 4.0 (I-4.0) technologies have created tremendous disruptions to the financial services industry, as they also have done in manufacturing and other industries. These digital technologies are designed to improve production efficiencies, the key competitive advantage to survive the modern, globalized markets. The combination of efficiency-driven property and digital structure makes I-4.0 technologies promising change agents for sustainability [1-7] and sustainable development [8-10]. They eliminate processes and resources that have caused irreversible environmental problems, resource depletion, and ecological imbalances [1]. To that effect, governments worldwide have acknowledged that I-4.0 technologies, including financial technology (FinTech), must be leveraged to recuperate environmental sustainability by de-materializing production and consumption, resulting in significantly lower use of natural resources [2-3,5]. The establishment of the United Nations Secretary General’s Special Advocate for Inclusive Finance for Development (UNSGSA) [4] affirms the contributions of FinTech financial services (henceforth, FinTech) to sustainable development [9-10]. FinTech contributes to financial inclusion by providing unbanked and underbanked consumers, especially low-income households and minority groups, access to affordable and convenient financing to help increase their economic opportunities [8,4]. FinTech financial services reduce costs, enhance the quality of financial services, increase employment rates, reduce poverty by lowering transaction costs, facilitate everyday personal and professional life [5-6], and provide financial access through microfinance and crowdfunding [5]. Consumers’ digital literacy and skills would also be enhanced through technology in financial services. FinTech financial services reduce energy consumption (e.g., fuel) and increase the protection of the environment (e.g., carbon emission) [1,6,7]. While conceptually solid, in reality, these benefits cannot be realized because FinTech financial services adoption is low.

While realizing the impact on sustainability requires FinTech natives (sustainable and mass adoption), consumers are still reluctant to embrace FinTech due to its controversies [11-16]. Consumers seem to consider the dangers and risks triggered by FinTech as more consequential than its benefits, which include conveniences, monetary savings, fast and seamless transactions, and economic efficiency [11-12,17-20]. FinTech is linked to cyber-related risks broadly categorized into loss of privacy, compromised data security, rising financial losses due to frauds and scams, unclear legal status, lack of regulations, and risks that FinTech providers lack operational effectiveness [13]. Most of these risks are caused by the misuse and abuse of data, which has become more accessible in the digital universe. Despite the controversies, [16-19] argued that previous studies on FinTech have focused on its benefits. They [16-19] attributed the problem to the over-reliance on the popular technology acceptance theories such as the Technology Acceptance Model (TAM), Theory of Planned Behavior (TPB), Diffusion of Innovations theory (DOI), and Unified Theory of Acceptance and Use of Technologies (UTAUT). Focusing on benefits could lead to suboptimal findings since the ubiquitous use of FinTech involves a complex trade-off between perceived benefits (returns/gains) and perceived risk (losses). It also does not justify that the cause of most financial problems bank consumers face in recent times is the failure to anticipate and manage risks and uncertainties [21]. This study addresses the gap in the literature by proposing the multidimensional benefit-risk perceptions in the Net Valence Framework (NVF) to explain FinTech behavioral adoption during the pandemic.

To a great extent, the COVID-19 pandemic has presented a solid inducement for migrating to FinTech (and other I-4.0 technologies) [18,22-27]. Powered by financial technology (the origin of the term “FinTech”) such as blockchain, big data, machine learning, and artificial intelligence (AI), FinTech has made it possible for consumers to execute financial transactions without the need for the physical presence of humans, money, or infrastructure. FinTech and I-4.0 technologies provide digital solutions to affected individuals, companies, and governments, thereby preventing the global economy from sinking into its worst depression. At the height of the COVID-19 pandemic, the government’s responses to contain physical movement and promote safe physical contact have created a massive adoption of FinTech [18,27]. Without disqualifying the role of rules and regulation in shaping behavior, there are sufficient reasons to argue that users’ behavioral shifts also drive FinTech adoption during the pandemic [28]. COVID-19 is a highly contagious acute respiratory virus (SARS-CoV-2) transmittable through physical contact with infected humans or objects, including banknotes and coins [26,29]. Studies by [23-25] have shown that the fear of becoming infected has changed consumer spending and purchasing behavior toward online platforms. Before the vaccines were available, an infection could lead to fatal consequences and caused many to suffer from morbid [25,28] and comorbid disorders [26]. In the FinTech context, As explained by [18], consumers suppress their concerns over FinTech risks to avoid COVID-19 infection. However, had [18] explicitly examined the fear of COVID-19, the result would have proven the role of consumer sentiment in technology adoption. The present study addresses the gap in the literature by proposing fear of COVID-19 to explain FinTech behavioral adoption during the pandemic.

Malaysia provides an excellent context for this study because the landscape of its FinTech services industry is built on a cash-dominated economy. Its FinTech industry is characterized as a (regulated) open competition ground for incumbents and FinTech startups (non-bank and digital banks). Before the COVID-19 outbreak, FinTech adoption in Malaysia was slow, despite its plan to transform the country into a cashless society [30]. The transformation is expected to result in cost savings worth 1% of the country’s Gross Domestic Product (GDP) [31-32]. Malaysia projected that technology-based innovations, specifically FinTech, would generate desired outcomes of financial inclusion; (i) convenient accessibility, (ii) high uptakes, (iii) responsible usage, and (iv) high satisfaction [32]. In their latest Financial Sector Blueprint 2022-2026 [31], Bank Negara Malaysia (BNM, the central bank of Malaysia) revised the e-payment per capita a compounded annual growth rate (CAGR) to more than 15% to speed up the digital transformation plan. At the same time, the country has invested in various initiatives to circumvent the FinTech threats. In 2020, the Malaysia Computer Emergency Response Team (MyCERT)[1] recorded 10,790 cyber security incidents, up from 10,722 in 2019. MyCERT projects that the potential loss due to incidents from 2020 to 2024 could amount to MYR51 billion, four times the expected cost savings if its digital transformation plan materialized [31]. The significant potential for monetary loss and the dynamism of its FinTech industry present Malaysia as a good setting to examine how perceived risks influence FinTech adoption during times of crisis when FinTech solutions are most critical.

In light of the tremendous FinTech financial service uptakes due to COVID-19, the increasing cybercrime incidents, and the burgeoning sustainability issues, this study is a timely attempt to close the gaps in the literature by considering the antecedents of FinTech adoption from the perspectives of NVF [33], Protection Motivation Theory (PMT) [34], and Sustainable Information Society (SIS) theory [36]. While most previous studies on FinTech services examine determinants of intention to adopt FinTech services [17-19,36-39], the present study focuses on the experience of adopting FinTech during the pandemic, which corroborates the goal to change the pandemic-induced behavior to loyalty [40]. This study addresses these gaps in the FinTech adoption literature through the following objectives:

  • To examine whether perceived benefit and risk significantly influence FinTech behavioral adoption;
  • To examine whether bank consumers’ fear of COVID-19 moderates the relationship between perceived benefit and risk with FinTech behavioral adoption;
  • To examine whether FinTech behavioral adoption contributes to sustainability.

Overall, this study contributes to the literature in the following ways. First, it addresses the gap in the literature on the impact of FinTech adoption on sustainability from the bank consumer perspective [41]. Although the COVID-19 pandemic has accelerated FinTech adoption, its implications for sustainability require the creation of FinTech natives. Second, it offers a context of a country undergoing a digital transformation while facing increasing cyber security incidents, therefore suitable to address the lack of empirical evidence on the effects of perceived risks [16-19]. As re-illustrated in Figure A1 (Appendix), Malaysia’s FinTech services industry is moving with the worldwide trend, reporting USD27.75 billion in 2021 and expecting a CAGR of 22% to reach USD74.38 billion by 2026. Earlier, [42] investigated FinTech adoption in Malaysia. However, their investigation of perceived risk was limited to internet banking, which is considered less complex and less risky than the recent FinTech services [12]. Third, this study takes a new twist in investigating the impact of the COVID-19 pandemic by proposing the role of fear of COVID-19 in influencing FinTech behavioral adoption. Fear of COVID-19 is a significant issue in the medical and psychiatric literature as it has caused comorbid [26] and morbid disorders [25,28]. It has received a lot of attention in financial market behavior studies, but this study is perhaps the first to examine its role in predicting technology adoption explicitly.

This study achieves its objectives by conducting an online survey on bank consumers between 4 December 2021 and 14 February 2022. As depicted in Figure A1 (Appendix), digital payments represent the largest segment (54%) in the Malaysian FinTech market, with a total transaction value of USD15.06 billion in 2021. Owing to its dominance, this study targets respondents among bank consumers who have experienced using FinTech to make payments or transfers through: i) online banking, ii) mobile banking, iii) contactless debit/credit/prepaid cards, iv) e-wallets, v) online foreign exchange, and/or vii) cryptocurrency e-wallets. These approaches yield 1,279 usable questionnaires that provide data from 400 randomly selected questionnaires. The data are tested using Structural Equation Modeling (SEM). The results of this study are relevant to banks, the incumbents of the financial services industry that is being disrupted by FinTech startups. For the FinTech companies, the results could help improve their FinTech adoption models, increasing their appeals to consumers. From the Malaysian policymaker’s perspective, the results can help formulate the most effective strategies to accelerate FinTech adoption and realize the country’s aspiration to become a cashless society and a regional leader in the digital economy.

2. Literature review and theoretical section

Unfortunately, I could not see relevant and compelling improvements in the theoretical framework, which still presents the major shortcomings that affected the original submission of this manuscript. In my own perspective, the authors are unable to build a fully-fledged and effective conceptual background in light of the references to the scientific literature they make. Rather, they seem to cherry-pick insights from previous scientific contributions and use them to work out several assumptions about the implications of big data governance and big data management on organizational innovativeness.

- I miss the conceptual nuances that allow us to understand how these phenomena are related. Of course, these nuances should be framed in lights of the theoretical background on which this paper is established. I also recommend the authors to adopt a more scientific style in arguing their research hypotheses. I rooted in the extant scientific debate, otherwise the study may be argued to suffer from limited conceptual reliability and consistency.

Authors’ Response: Thank you but we have explained the theoretical arguments in the literature review and the conceptual framework is developed from integration of three theories, as depicted clearly in Figure 1.

Also, about the implication of “big data governance and big data management on organizational innovativeness” – this comment seems mistaken from other papers.

3. METHODS

The methodology now includes some additional insights, which may help the reader assess this study's dependability. However, these revisions are insufficient to support the study's consistency and dependability. The authors are still unable to argue the motivations that triggered them to focus on the  particularly industry.

Authors’ Response: Thank you but these issues were never mentioned in earlier review reports. Still, we conducted various tests on our instrument and then data to ensure consistency and dependability (reliability). However, the argument about “industry” is unexpected because FinTech which is used to refer to financial services powered by financial technologies, are naturally meant for the financial service industry, which banks are the main player and incumbents. This is in addition to the justifications we provide in targeting bank consumers as our respondents.
